# Haste Makes Waste: A Simple Approach for Scaling Graph Neural Networks

**Rui Xue**[1]  **Tong Zhao**[2]  **Neil Shah**[2]  **Xiaorui Liu**[3]

## Abstract

Graph neural networks (GNNs) have demonstrated remarkable success in graph representation learning and various sampling approaches have been proposed to scale GNNs to applications with large-scale graphs. A class of promising GNN training algorithms take advantage of historical embeddings to reduce the computation and memory cost while maintaining the model expressiveness of GNNs. However, they incur significant computation bias due to the stale feature history. In this paper, we provide a comprehensive analysis of their staleness and inferior performance on large-scale problems. Motivated by our discoveries, we propose a simple yet highly effective training algorithm (REST) to effectively reduce feature staleness, which leads to significantly improved performance and convergence across varying batch sizes, especially when staleness is predominant. The proposed algorithm seamlessly integrates with existing solutions, boasting easy implementation, while comprehensive experiments underscore its superior performance and efficiency on large-scale benchmarks. Specifically, our improvements to state-of-the-art historical embedding methods result in a 2.7% and 3.6% performance enhancement on the ogbn-papers100M and ogbn-products dataset respectively, accompanied by notably accelerated convergence. The code can be found at https://github.com/RXPHD/REST.

## 1. Introduction and Related Works

Graph neural networks (GNNs) have emerged as powerful tools for representation learning on graph-structured data (Hamilton, 2020). They exhibit significant advantages across various general graph learning tasks, including node classification, link prediction, and graph classification (Kipf & Welling, 2016; Gasteiger et al., 2019; Veličković et al., 2017; Wu et al., 2019; Xue et al., 2023b). GNNs have also proven to be highly effective in various applications, such as recommendation systems, biological molecules, and transportation (Tang et al., 2020; Sankar et al., 2021; Fout et al., 2017; Wu et al., 2022; Zhang et al., 2024). However, scalability becomes a dominant bottleneck when applying GNNs to large-scale graphs. This is because the recursive feature propagations in GNNs lead to the notorious neighborhood explosion problem since the number of neighbors involved in the mini-batch computation grows exponentially with the number of GNN layers (Hamilton et al., 2017; Chen et al., 2018; Han et al., 2023). This is particularly undesirable for deeper GNNs that try to capture long-range dependency on large graphs using more feature propagation layers. This neighborhood explosion problem poses a significant challenge as it cannot be accommodated within the limited GPU memory and computation resources during training and inference, which hampers the expressive power of GNNs and limits their applications to large-scale graphs.

Existing works have made significant contributions to address this issue from various perspectives, such as sampling, distributed computing, and pre-computing or post-computing approaches. In this paper, we focus on the most commonly used sampling approaches. In particular, various sampling approaches have been proposed to mitigate the neighborhood explosion problem in large-scale GNNs (Hamilton et al., 2017; Chen et al., 2018; Chiang et al., 2019; Zeng et al., 2020). These sampling approaches attempt to reduce the graph size by sampling nodes and edges to lower the memory and computation costs in each mini-batch iteration. However, they also introduce estimation variance of embedding approximation in the sampling process, and they inevitably lose accurate graph information.

Historical embedding methods have emerged as promising solutions to address this issue, such as VR-GCN (Chen et al., 2017), MVS-GCN (Cong et al., 2020), GAS (Fey et al., 2021), and GraphFM (Yu et al., 2022). They propose to reduce the estimation variance of sampling methods using historical embeddings. Specifically, during each training iteration, they preserve intermediate node embeddings at

---

[1]Department of Electrical and Computer Engineering, North Carolina State University, Raleigh, US [2]Snap Inc., Bellevue, US [3]Department of Computer Science, North Carolina State University, Raleigh, US. Correspondence to: Xiaorui Liu <xliu96@ncsu.edu>.

*Proceedings of the 42nd International Conference on Machine Learning*, Vancouver, Canada. PMLR 267, 2025. Copyright 2025 by the author(s).

each GNN layer as historical embeddings, which are then used to reduce estimation variance in future iterations. The historical embeddings can be stored offline on CPU memory or disk and will be used to fill in the inter-dependency from nodes out of the current mini-batch. Therefore, it does not ignore any nodes or edges, which reduces variance and keeps the expressivity properties of the backbone GNNs with strong scalability and efficiency. Please refer Appendix M for a detailed summary of related works.

Despite their promising scalability and efficiency, in this paper, we discover that all historical embedding methods *can not* consistently outperform vanilla sampling methods such as GraphSAGE which do not utilize historical embeddings, especially on larger graph datasets. Moreover, their prediction performance and convergence suffer from significant degradation when the batch size decreases. Although some existing works have attempted basic analyses (Huang et al., 2023; Wang et al., 2024), these efforts are quite limited, and the underlying reasons remain unclear. In this paper, we provide the first and most comprehensive analysis, incorporating both theoretical and empirical studies, to reveal the emergence of significant embedding staleness with these methods and its severe negative impact on GNN training. Our preliminary study reveals that the primary obstacle lies in the fact that these stored historical embeddings are not computed by the most recent model parameters, leading to a phenomenon known as *staleness*. Staleness represents the approximation error between the true embeddings computed using the most recent model parameters and the stale embeddings cached in the memory. This issue is pervasive across all historical embedding methods and significantly impacts training convergence and model performance, particularly when models undergo frequent updates—such as training GNNs on large-scale graphs with smaller batch sizes. Due to the substantial bias introduced by stale embeddings, these approaches cannot fully realize their potential in performance, despite their excellent efficiency and scalability.

Motivated by our findings and analysis, we propose a simple yet highly effective solution to reduce feature staleness by decoupling the forward and backward phases and dynamically adjusts their execution frequencies, allowing the memory table to be updated more frequently than the model parameters. Our aim is to alleviate the current bottleneck on performance and convergence while preserving exceptional efficiency. Our proposed framework is highly flexible, orthogonal and compatible with any sampling methods, memory-based models, and various GNN backbones. Comprehensive experiments demonstrate its effectiveness in addressing the significant staleness issue present in large-scale datasets, leading to superior prediction performance and accelerated convergence while maintaining or even improving running time and memory efficiency. Specifically, our enhancements to state-of-the-art historical embedding

methods yield a 2.7% and 3.6% performance boost on the ogbn-papers100m and ogbn-products dataset respectively. Notably, these improvements are achieved with significantly faster convergence times without sacrificing efficiency.

## 2. Preliminary Study

In this section, we illustrate that the staleness of historical embeddings serves as the bottleneck for existing historical embedding approaches when handling large-scale graphs. We aim to support this claim with empirical studies on prediction performance, training convergence, memory persistence, and approximation errors.

### 2.1. Message Passing with Historical Embeddings

**Formulations.** Let $h_v^l$ represent the feature embedding of the in-batch node $v$ in $l$-th layer and $f_\theta^{(l)}$ denote the message passing update in a $l$-th layer with parameter $\theta$. The standard sampling method for a mini-batch $B \subset \mathcal{V}$ can be expressed as follows:

$$h_v^{(l+1)} = f_\theta^{(l+1)}(h_v^l, [h_u^l]_{u \in \mathcal{N}(v) \cap B}) \qquad (1)$$

where $\mathcal{N}(v) \cap B$ is the in-batch 1-hop neighborhood of in-batch node $v$. As discussed in the introduction, the sampling methods (e.g., GraphSAGE, FastGCN) drop all out-of-batch neighbors $\mathcal{N}(v) \backslash B$ and cannot aggregate their embeddings $[h_u^l]_{u \in \mathcal{N}(v) \backslash B}$, which results in high estimation variance. To address this issue, historical embedding methods use historical embeddings $[\bar{h}_u^l]_{u \in \mathcal{N}(v) \backslash B}$ to approximate $[h_u^l]_{u \in N(v) \backslash B}$. For example, the message passing in GAS (Fey et al., 2021) can be denoted as:

$$h_v^{(l+1)} = f_\theta^{(l+1)}(h_v^l, [h_u^l]_{u \in \mathcal{N}(v)}) \qquad (2)$$

$$= f_\theta^{(l+1)}(h_v^l, \underbrace{[h_u^l]_{u \in \mathcal{N}(v) \cap B}}_{\text{in-batch neighbors}} \cup \underbrace{[h_u^l]_{u \in \mathcal{N}(v) \backslash B}}_{\text{out-of-batch neighbors}}) \qquad (3)$$

$$\approx f_\theta^{(l+1)}(h_v^l, \underbrace{[h_u^l]_{u \in \mathcal{N}(v) \cap B}}_{\text{in-batch neighbors}} \cup \underbrace{[\bar{h}_u^l]_{u \in \mathcal{N}(v) \backslash B}}_{\text{historical embeddings}}), \qquad (4)$$

followed by feature memory update $\bar{h}_v^{l+1} = h_v^{l+1}$. It is evident that historical embedding methods effectively reduce the estimation variance of $h_v^{(l+1)}$ without further expansion on the neighbor size. An illustrative example is presented in Figure 1 to elaborate on these concepts. However, these historical embedding methods also incur larger estimation bias due to the approximation using historical embeddings.

### 2.2. Empirical Study

**Inferior performance.** While historical embedding methods exhibit both strong performance and scalability on small-

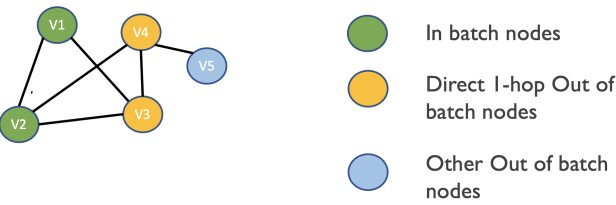

Figure 1: We sample a batch with a batch size of 2 (e.g., nodes 1 and 2), their direct one-hop neighbors are nodes 3 and 4. Nodes 1 and 2 are considered "in-batch" nodes, as they always use real-time updated embeddings and participate in model updates with gradient computation; Nodes 3 and 4 are considered "direct one hop out-of-batch" nodes (simply referred to as "out-of-batch" nodes (Yu et al., 2022)) because they are approximated using historical embeddings and their embeddings are not real-time updated; Node 5 is called "other out-of-batch" node that is not used in the current iteration.

scale and medium-scale graph datasets, they can not consistently outperform simpler variants such as GraphSAGE, Cluster-GCN, and GraphSAINT on larger datasets such as Reddit and ogbn-products as shown in multiple works (Fey et al., 2021; Yu et al., 2022; Xue et al., 2023a). The bitter truth is that their prediction performance can be significantly inferior in many cases even though they cost much more offline memory to store hidden features in all intermediate layers. For example, as one of the state-of-the-art historical embedding methods, GAS (GCNII) achieves only 77.20% accuracy on ogbn-products while GraphSAGE and GraphSAINT achieve 78.70% and 79.08%. The potential of historical embedding methods is significantly limited. However, their fundamental limitations of inferior performance are yet unclear. Next, we will provide a comprehensive analysis from the feature staleness perspective.

**Convergence analysis.** We present a deeper empirical convergence analysis to reveal the performance bottleneck of historical embedding methods. To underscore this point, we present convergence curves comparing GraphSAGE, two representative historical embedding methods, GAS and GraphFM with GCN as backbone. We explore the impact of staleness on accuracy and convergence by using datasets of different scales (ogbn-arxiv and ogbn-products) and experimenting with both small and large batch sizes. Comparisons are conducted under identical hyperparameter settings. "Small" and "Large" denote the small and large batch sizes settings, respectively. In particular, when using small batch sizes or larger graphs, model updates occur more frequently within a single epoch. Consequently, staleness becomes significant and exerts a dominant influence on performance.

From Figure 2, when using the small ogbn-arxiv dataset with a larger batch size (staleness is minor), GraphSAGE exhibits slower convergence and poorer performance than

GAS; however, in all other cases (when staleness is large), GraphSAGE converges more rapidly and achieves better performance. Despite GraphFM's utilization of current one hop neighbors to mitigate staleness, its impact is limited. This observation supports our earlier conclusion that historical embedding methods perform well when staleness is not dominant but face limitations in realizing their full potential under conditions of high staleness.

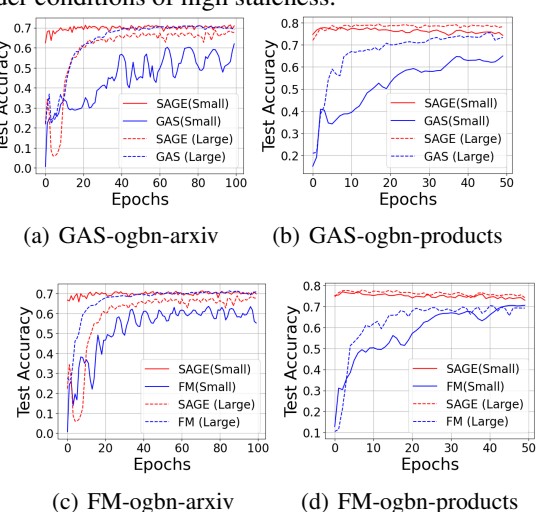

(a) GAS-ogbn-arxiv     (b) GAS-ogbn-products

(c) FM-ogbn-arxiv     (d) FM-ogbn-products

Figure 2: GAS and FM exhibit inferior performance and slower convergence, especially on larger datasets (i.e., ogbn-products) or small batch size (i.e., "Small").

**Memory persistence.** To clearly illustrate staleness, we introduce a new notation of "memory persistence" for each node embedding, which quantifies the duration that a historical embedding remains in memory before being updated, measured in the number of optimizer steps. A large memory persistency means it has not been updated frequently, which may lead to stronger feature staleness. We present the average memory persistence over all nodes in GAS (Fey et al., 2021) on ogbn-arxiv and ogbn-products datasets with varying batch sizes in Figure 3. It can be observed that smaller batch size incurs larger memory persistence. In particular, when employing a batch size of 1024 on ogbn-products, the model experiences around 2400 updates before updating the historical embedding, which results in a considerable persistence of stale features.

**Embedding approximation errors.** We present Figure 4 to show the approximation error of the embeddings between GAS and full batch $||\tilde{h}_v^{(L)} - h_v^{(L)}||$ on ogbn-arxiv and ogbn-products. Please refer Appendix J for GraphFM.

As shown in Figure 4, the embeddings in the final layer gradually diverge from the true embeddings computed using full-batch data as the model parameters are updated through stochastic gradient descent. Consequently, the approximation errors accumulate and grow over epochs due to the accumulated staleness of the historical embeddings. These empirical analyses demonstrate that historical embedding

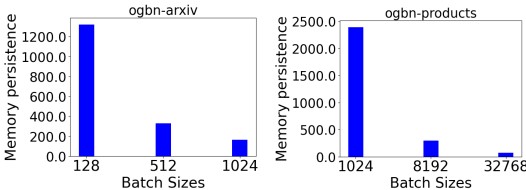

Figure 3: Embedding Memory persistence (GAS).

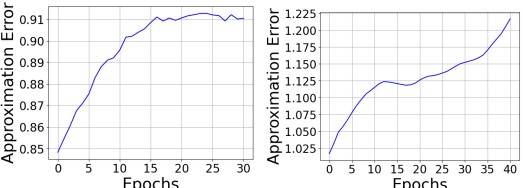

Figure 4: Approx. Error on Arxiv(L) and Products(R).

methods such as GAS are significantly influenced by the quality of the historical embeddings utilized for message passing.

## 3. Methodology

In this section, we first elucidate the primary factor influencing the final performance through theoretical analysis. Subsequently, we introduce a simple yet novel and effective approach designed to mitigate staleness at its source, thereby significantly enhancing performance. Furthermore, we present an advanced version that takes into account the importance of nodes, demonstrating improved robustness and performance, particularly in scenarios where staleness is pronounced.

### 3.1. Approximation Error Analysis

While a historical embedding approaches efficiently preserve the expressive power of the original GNNs and reduce variance, they introduce approximation errors between the exact embeddings and historical embeddings. This limitation becomes more pronounced, particularly in scenarios with large graphs or when employing deep GNN models. To illustrate this challenge and provide motivation for our work, we first introduce a theory to demonstrates that the approximation error of the gradient can be upper-bounded by the accumulation of staleness in each layer. We adhere to the assumptions used in existing works (Fey et al., 2021).

**Theorem 3.1** (Gradient Approximation Error). *Consider a L-layers GNN $f_\theta^{(l)}(h)$ with Lipschitz constant $\alpha^{(l)}$, $UPDATE_\theta^{(l)}$ function with Lipschitz constant $\beta$, $l = 1, \ldots, L$. $\nabla L_\theta$ has Lipschitz constant $\varepsilon$. If $\forall v \in V$, $||\bar{h}^{(l)} - h^{(l)}||$ denotes the staleness between historical embeddings and true embeddings from full batch aggregations, then the approximation error of gradients is bounded by:*

$$||\nabla L_\theta(\tilde{h}_v^{(L)}) - \nabla L_\theta(h_v^{(L)})|| \tag{5}$$

$$\leq \varepsilon \sum_{l=1}^{L} \left( \prod_{k=l+1}^{L} \alpha^{(k)} \beta |\mathcal{N}(v)| * ||\tilde{L}_v,|| * ||\bar{h}^{(k-1)} - h^{(k-1)}|| \right).$$

The proof of the above theorem and the explanation of the assumption can be found in Appendix A. We can conclude that the gradient is influenced by the number of layers $L$, the number of neighbors $\mathcal{N}(v)$, and the per-layer approximation error $||\bar{h}^{(k-1)} - h^{(k-1)}||$. If we effectively reduce the

approximation error, we can efficiently minimize this error accumulation across layers and tighten the upper bound.

### 3.2. REST: REducing STaleness

As discussed in Appendix M, existing works such as GAS (Fey et al., 2021), GraphFM-OB (Yu et al., 2022), and ReFresh (Huang et al., 2023) endeavor to tackle staleness from various angles, such as minimizing nodes interconnectivity, applying regularization techniques, employing feature momentum by in-batch nodes, and expanding in-batch neighbor size, to alleviate staleness. However, these approaches can not control the staleness effectively and efficiently. Consequently, they typically yield only marginal performance improvements. We attribute this to their inability to resolve the staleness issue stemming from the disparity in updating frequency of model parameters and the memory tables for historical embeddings. Therefore, we propose a simple yet novel approach to adjust the frequency of forward and backward propagation, addressing this problem at its root. Note that our model is orthogonal to all of the existing techniques and can be seamlessly integrated.

In standard training procedures, forward and backward propagation are typically interconnected. During forward propagation, intermediate values are computed at each layer, which are then used during backward propagation to compute gradients and update the model parameters. This coupling ensures that the gradients computed during backward propagation accurately reflect the effect of the model parameters on the loss function, allowing for effective parameter updates during optimization. Under these limitations, it is natural to mitigate feature staleness by using larger batch sizes, but it will cost higher memory and computation costs.

**Effortless and Effective Staleness Reduction.** In historical embedding methods, in addition to the computation of intermediate values, forward propagation also plays a role in updating the memory table by in-batch nodes. Consequently, only the historical embeddings of the in-batch nodes are updated during a single training iteration. Taking GAS as an example, the entire table completes an update after $k = \frac{n}{B}$ training iterations, where $n$ is the number of nodes and $B$ is the batch size. However, during this period, model parameters undergo updates $k$ times. The inability to refresh historical embeddings using the most recent parameters results in the approximation errors and it accumulates

at each training iteration. When retrieving the embedding from memory, this error introduces significant bias to the final representation, as illustrated in Figure 4. The root issue lies in the fact that the frequency of updating the memory table is too slow compared with the frequency of model updates. The disparity between the frequency originates from the forward and backward propagation are tightly coupled.

Building upon this analysis, we propose to simply decouple the forward and backward propagations and adjust their execution frequency. The process is illustrated in Figure 5 and outlined in Algorithm 1. Specifically, $F$ times the forward propagations on batch $\mathbf{B}_1 \ldots \mathbf{B}_F$ are initially executed without requiring gradient (line 7). This process updates the memory table at each layer using in-batch nodes exclusively. Subsequently, another forward propagation with different batch $\tilde{\mathbf{B}}_j$ is executed with requiring gradients for the following model update by backward (line 10). The updated embeddings from every forward propagation (line 11 and line 8) are then cached in memory to facilitate the updating of embeddings and mitigate staleness issues. Hence, our REST approach effectively reduces the gap in update frequencies from $k$ to $\frac{k}{F}$. The hyperparameter $F$ directly governs the staleness and allows for flexible adjustments without increasing memory cost. Our experiments in Appendix D empirically demonstrate that employing $F = 1$ not only yields significantly improved performance compared to baselines but also accelerates convergence than baselines. This training process is very simple and flexible since forward passes are relatively cheap and trivially parallelizable. For example, it can be further accelerated by parallelizing the forward propagations (line 7 and line 10) on multiple GPUs. However, in this paper, we only use single GPU for a fair comparison with the baselines.

---

**Algorithm 1** REST Technique

---

1: **Input:** Input graph $\mathcal{G} = (\mathcal{V}, \mathcal{E})$, GNN $f(\mathbf{X}, \Theta^0)$
2: **Output:** Fine-tuned GNN $\tilde{f}(\mathbf{X}, \Theta^*)$
3: **Begin**
4: **for** each mini-batch $\mathbf{B}_1 \ldots \mathbf{B}_F$;
5: each mini-batch $\tilde{\mathbf{B}}_j$ **do**
6:     **for** $i$ in updating frequency $F$ **do**
7:         $\mathbf{H}_1 = f(\mathbf{B}_i, \Theta^k)$: offline forward propagation
8:         Cache into memory $\mathbf{M} \leftarrow \mathbf{H}_1[\text{in-batch}]$
9:     **end for**
10:    $\mathbf{H}_2 = f(\tilde{\mathbf{B}}_j, \Theta^k)$: forward propagation with backward
11:    Cache into memory $\mathbf{M} \leftarrow \mathbf{H}_2[\text{in-batch}]$
12:    Compute loss and gradient update
13: **end for**

---

Finally, we can prove that REST achieves a faster convergence rate theoretically (Chen et al., 2017; Shi et al., 2023):

**Theorem 3.2.** *Given the upper bound of the expectation of gradients' norm in the state-of-the-art historical embed-*

dings methods such as GAS, LMC, and REST, which is

$$E[||\nabla_\theta L(\theta_R)||_2]$$
$$\leq \left( \frac{2(L(\theta_1) - L_\theta^* + G_\theta)}{N^{\frac{1}{3}}} + \frac{\epsilon G_\theta}{N^{\frac{2}{3}}} + \frac{G_\theta}{N(1 - \sqrt{\rho})} \right)^{\frac{1}{2}} \tag{6}$$

*where $G_\theta$ is the upper bound of gradient approximation error, $N$ is the number of iterations, $R$ is chosen uniformly from $[N]$, $\rho \in (0, 1)$. Based on Theorem 3.1, the upper bound $G_\theta$ of REST is tighter than that of existing historical embedding methods. Consequently, the convergence speed of REST surpasses that of existing works.*

The detailed proof can be found in Appendix B. In addition to its simplicity, REST exhibits remarkable generality. Notably, REST can also address staleness in backward by maintaining a historical gradient cache and updating it at a different frequency $\tilde{F}$ without updating model parameters (see ***Bidirectional REST*** in Section 4.5). Furthermore, for the application of REST to asynchronous updates, please refer to Appendix P for the discussion.

### 3.3. REST: Sampling Strategy

It's crucial to emphasize that our algorithm is versatile and applicable to any historical embedding methods. Furthermore, this flexibility also extends to the sampling strategy employed in forward propagation, encompassing various techniques such as uniform sampling, importance sampling, and any custom sampling method tailored to the specific requirements of the task at hand. Here, we propose a feasible option to further address the staleness issue by considering the varying significance of individual nodes within the graph, termed **REST-IS (Importance Sampling)**. Notably, nodes with high degrees are more likely to serve as neighbor nodes for the sampled nodes, making a more substantial contribution to the final representation and carrying higher importance. The staleness in these highly important node embeddings makes a more detrimental impact on performance compared to nodes with lower importance.

Motivated by this potential issue, rather than employing conventional importance sampling techniques, we propose a novel and efficient method for importance sampling. Our method utilizes the neighbor nodes from batch ($\tilde{\mathbf{B}}_j$ in Algorithm 1) in the forward propagation for model updates (line 10) and lets them serve as the in-batch nodes in batch $\mathbf{B} = \mathbf{B}_1 \cup \mathbf{B}_2 \cdots \cup \mathbf{B}_F$ for updating the memory table (line 7). We are motivated by the fact that if one node is frequently served as neighbor nodes by different batches, it's likely to holds higher importance than other nodes in the graph. We summarize this approach by presenting formulas:

$$h_v^{(l+1)} = f_\theta^{(l+1)}(h_v^l, [h_u^l]_{u \in \mathcal{N}(v)}) \tag{7}$$

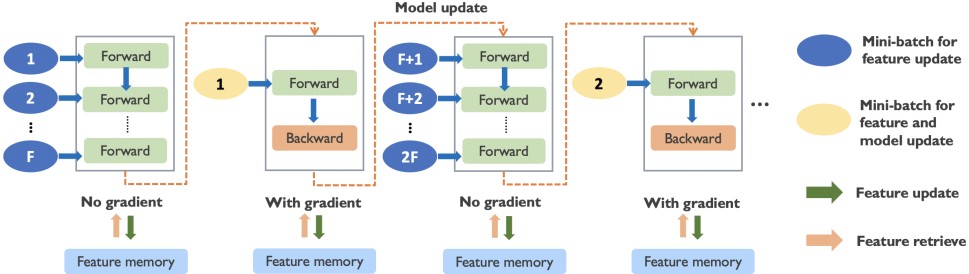

Figure 5: Training process for the proposed REST technique. (1) F mini-batches $\mathbf{B}_1 \ldots \mathbf{B}_F$ (blue ellipse) are executed without computing gradients to update memory table (2) Another one mini-batch $\tilde{B}$ (yellow ellipse) is processed with gradient computation to update the model parameters.

$$= f_\theta^{(l+1)}(h_v^l, [h_u^l]_{u \in \mathcal{N}(v) \cap \tilde{B}} \cup [h_u^l]_{u \in \mathcal{N}(v) \setminus \tilde{B}}) \quad (8)$$

$$\approx f_\theta^{(l+1)}\Bigg(h_v^l, [h_u^l]_{u \in \mathcal{N}(v) \cap \tilde{B}}$$

$$\cup [f_\theta^{(l+1)}(h_w^l, [h_w^l]_{w \in \mathcal{N}(u) \cap B} \cup \underbrace{[\bar{h}_w^l]_{w \in \mathcal{N}(u) \setminus B}}_{\text{Historical}})]_{u \in \mathcal{N}(v) \setminus \tilde{B}}\Bigg)$$

$$(9)$$

Similar to the initial approach, we employ historical embeddings $[\bar{h}_w^l]_{w \in \mathcal{N}(u) \setminus B}$ for these out-of-batch nodes. Subsequently, we store both the in-batch nodes $[h_w^l]_{w \in \mathcal{N}(u) \cap \tilde{B}}$ and the in-batch nodes $[h_u^l]_{u \in \mathcal{N}(v) \cap B}$ into memory for updating historical embeddings, as they are all freshly computed. As only neighbor nodes are chosen as in-batch nodes during memory table update, this approach simply and efficiently accounts for the importance of each node. This approach can be especially helpful when dealing with a high staleness situation, as shown in our experiments.

## 4. Experiments

In this section, we conduct experiments to showcase the superior ability of our proposed algorithms in improving the performance while accelerating the convergence, especially when staleness is predominant.

**Experimental setting.** We first provide a best performance comparison with multiple major baselines including GCN (Kipf & Welling, 2016), GraphSage (Hamilton et al., 2017), FastGCN (Chen et al., 2018), LADIES (Zou et al., 2019), Cluster- GCN (Chiang et al., 2019), Graph-SAINT (Zeng et al., 2020), SGC (Wu et al., 2019), GN-NAutoScale(GAS) (Fey et al., 2021), VR-GCN (Chen et al., 2017), MVS-GCN (Cong et al., 2020) and GrpahFM (Yu et al., 2022) on three widely used large-scale graph datasets including REDDIT, ogbn-arxiv, and ogbn-products (Hu et al., 2020). Based on our current computation resources, we also include two larger datasets, ogbn-papers100m and MAG240M, in Appendix E. For our model, we opt to use GAS based approach for simplicity since it's proven the best (Fey et al., 2021). We provide comparison with recent related baselines Refresh (Huang et al., 2023) and LMC (Shi et al., 2023) in Appendix G. We also include comparisons with several minor baselines (Wang et al., 2024; Bai

et al., 2023) in Appendix O, which have not yet been officially published or do not have publicly available code, ensuring that our results are as comprehensive as possible. The hyperparameter tuning of baselines closely follows their default settings (See Appendix R). Besides, we have chosen to stick to the baseline settings (such as sampling methods) in this work to maintain consistency across our baselines although numerous sophisticated tricks could potentially enhance performance. We emphasize that our proposed method is orthogonal to them. For our model, we choose a frequency of $F = 1$, as it has been proven to be effective in ablation study. The performance is shown in Table 1. In instances where performance metrics or hyperparameters are not reported, we denote them with "—" to align with the baseline papers. OOM stands for out-of-memory. Note that, REST uses default uniform sampling while REST-IS uses proposed importance sampling.

### 4.1. Performance

Table 1 showcases the accuracy comparison of REST and REST-IS with state-of-the-art baseline methods across multiple backbones. We can observe the following:

• All historical embedding methods exhibit inferior performance compared to simple models like GraphSAGE on the large dataset, ogbn-products, aligning with our earlier observations. Note that GraphFM uses in-batch nodes to compensate for staleness to a certain extent. However, it only achieves marginal improvement. This stems from their inability to eliminate staleness at its source, as discussed in the Section 3.2, whereas our approach can.

• Compare with all other baselines on large scale datasets, our proposed algorithms outperform all baselines on ogbn-arxiv and ogbn-products and reach comparable performance on Reddit. Specifically, REST can achieve 73.2% on ogbn-arixv and 80.5% on ogbn-products. Note that, it boosts the performance of GAS by 3.6% when using APPNP as GNN backbone. The only exceptions are that GraphSAINT slightly outperforms our method. However, they require complicated and time-consuming preprocessing.

• When combined with results from larger datasets such as ogbn-products, ogbn-papers100M, and MAG240M, which exhibit significant staleness, our model demonstrates a **substantial performance boost** compared to smaller datasets (e.g., ogbn-arxiv) (see Appendix E). This clearly indicates that our approach effectively addresses the staleness issue. Given the increasing size of datasets in practical, these results highlight the potential and necessity of our work.

Table 1: Accuracy comparison (%) with major baselines. Cyan denotes the best performance under a specific dataset.

| Framework | GNNs | # nodes
230K
# edges
11.6M
REDDIT | # nodes
169K
# edges
1.2M
ARXIV | # nodes
2.4M
# edges
61.9M
PRODUCTS |
|---|---|---|---|---|
| **Major** | GraphSAGE | 95.4± 0.2 | 71.5± 0.2 | 78.7± 0.1 |
| | FastGCN | 93.7± 0.1 | — | — |
| | LADIES | 92.8± 0.2 | — | — |
| | Cluster-GCN | 96.6± 0.1 | — | 79.0± 0.2 |
| | GraphSAINT | 97.0± 0.1 | — | 79.1± 0.1 |
| | SGC | 96.4± 0.1 | — | — |
| | VR-GCN | 94.1± 0.1 | 71.6± 0.1 | 76.4± 0.1 |
| | MVS-GNN | 94.9± 0.1 | 71.6± 0.1 | 76.9± 0.1 |
| **Full Batch** | GCN | 95.4± 0.2 | 71.6± 0.1 | OOM |
| | APPNP | 96.1± 0.1 | 71.8± 0.1 | OOM |
| | GCNII | 96.1± 0.1 | 72.8± 0.2 | OOM |
| **GAS** | GCN | 95.4± 0.1 | 71.7± 0.2 | 76.7± 0.2 |
| | APPNP | 96.0± 0.1 | 71.9± 0.2 | 76.9± 0.1 |
| | GCNII | 96.7± 0.1 | 72.8± 0.1 | 77.2± 0.3 |
| **GraphFM** | GCN | 95.4± 0.2 | 71.8± 0.2 | 76.8± 0.2 |
| | GCNII | 96.8± 0.1 | 72.9± 0.1 | 77.4± 0.3 |
| **REST (Ours)** | GCN | 95.6± 0.1 | 72.2± 0.2 | 79.6± 0.1 |
| | APPNP | 96.4± 0.1 | 72.4± 0.1 | 80.0± 0.1 |
| | GCNII | 96.8± 0.1 | 73.2± 0.1 | 79.8± 0.2 |
| **REST-IS (Our)** | GCN | 95.7± 0.1 | 72.3± 0.1 | 78.6± 0.1 |
| | APPNP | 96.5± 0.1 | 72.4± 0.2 | 80.5± 0.2 |
| | GCNII | 96.8± 0.1 | 72.8± 0.2 | 79.6± 0.3 |

### 4.2. Improvement for Historical Embeddings

**Experimental setting.** According to our analysis in methodology, our proposed approach can efficiently reduce the staleness in all of the memory-based algorithms, especially when the batch size is small. Hence, we conduct further experiments under different batch sizes to show the effectiveness of our algorithm. We first present a comparison with three representative historical embedding methods, GNAutoScale (GAS) (Fey et al., 2021), VR-GCN (Chen et al., 2017) and MVS-GCN (Cong et al., 2020). The results are shown in Table 2 and Table 3. In these experiments, we closely adhere to their settings, including the datasets they utilized in their paper and official repositories.

In detail, both VR-GCN and MVS-GCN leverage neighbor sampling, akin to the approach introduced by GraphSAGE (Hamilton et al., 2017), whereas GAS adopts a different strategy. GAS begins the process by employing the METIS algorithm to partition the graph into distinct clusters. Following this partitioning, one or more of these clusters are thoughtfully selected to constitute a batch for computation purposes. The total number of clusters is detailed in Table 2 under the name "parts". The number under "BS" (batch size) indicates the number of clusters included in one batch. **Performance Analysis.** From results, we can observe:

Table 2: Accuracy (%) improvement for GAS. Cyan denotes the best performance under a specific GNN model.

| DATASET | GNN | PARTS | BS | GAS | +REST | +REST-IS |
|---|---|---|---|---|---|---|
| **Products** | **GCN** | 70 | 5 | 75.6 ± 0.4 | 77.6 ± 0.2 | 77.9 ± 0.2 |
| | | | 10 | 76.5 ± 0.2 | 79.4 ± 0.2 | 78.6 ± 0.1 |
| | **APPNP** | 40 | 5 | 75.0 ± 0.4 | 79.7 ± 0.2 | 80.4 ± 0.1 |
| | | | 10 | 76.8 ± 0.1 | 80.1 ± 0.1 | 80.4 ± 0.1 |
| | **GCNII** | 150 | 5 | 74.8 ± 0.6 | 75.9 ± 0.3 | 76.2 ± 0.3 |
| | | | 20 | 76.9 ± 0.3 | 79.8 ± 0.3 | 79.6 ± 0.2 |
| **Reddit** | **GCN** | 200 | 20 | 94.8 ± 0.2 | 95.3 ± 0.1 | 95.4 ± 0.1 |
| | | | 100 | 95.4 ± 0.1 | 95.6 ± 0.1 | 95.7 ± 0.1 |
| | **APPNP** | 200 | 20 | 92.6 ± 0.2 | 95.9 ± 0.1 | 96.1 ± 0.1 |
| | | | 100 | 96.0 ± 0.1 | 96.4 ± 0.1 | 96.5 ± 0.1 |
| | **GCNII** | 200 | 20 | 93.9 ± 0.1 | 95.7 ± 0.1 | 95.7 ± 0.1 |
| | | | 100 | 96.7 ± 0.1 | 96.8 ± 0.1 | 96.8 ± 0.1 |
| **Arxiv** | **GCN** | 80 | 5 | 67.6 ± 0.6 | 71.1 ± 0.3 | 71.9 ± 0.1 |
| | | | 10 | 69.5 ± 0.5 | 71.3 ± 0.2 | 72.1 ± 0.1 |
| | | | 20 | 70.6 ± 0.2 | 71.5 ± 0.1 | 72.2 ± 0.2 |
| | | | 40 | 71.5 ± 0.2 | 72.2 ± 0.1 | 72.3 ± 0.2 |
| | **APPNP** | 40 | 5 | 69.3 ± 0.4 | 71.7 ± 0.2 | 72.3 ± 0.2 |
| | | | 10 | 70.0 ± 0.3 | 72.1 ± 0.2 | 72.4 ± 0.1 |
| | | | 20 | 71.6 ± 0.3 | 72.4 ± 0.1 | 72.3 ± 0.2 |
| | **GCNII** | 40 | 5 | 70.0 ± 0.3 | 72.6 ± 0.3 | 72.7 ± 0.2 |
| | | | 10 | 71.9 ± 0.2 | 72.7 ± 0.2 | 72.7 ± 0.1 |
| | | | 20 | 72.5 ± 0.3 | 73.1 ± 0.1 | 72.8 ± 0.2 |

Table 3: Improvement (%) for VR-GCN and MVS-GCN. Cyan and Magenta indicate the best performance for them.

| Dataset | Batch Size | VR-GCN | +REST | +REST-IS |
|---|---|---|---|---|
| **Products** | 10000 | 76.3 ± 0.3 | 77.4 ± 0.3 | 78.0 ± 0.2 |
| | 50000 | 76.4 ± 0.1 | 77.5 ± 0.1 | 78.1 ± 0.2 |
| **Reddit** | 256 | 91.7 ± 0.2 | 93.8 ± 0.2 | 94.1 ± 0.1 |
| | 512 | 93.0 ± 0.2 | 94.1 ± 0.2 | 94.2 ± 0.2 |
| | 2048 | 94.1 ± 0.1 | 94.5 ± 0.1 | 94.3 ± 0.1 |
| **Arxiv** | 128 | 70.4 ± 0.3 | 71.7 ± 0.3 | 71.9 ± 0.1 |
| | 512 | 71.5 ± 0.3 | 72.3 ± 0.2 | 72.4 ± 0.1 |
| | 2048 | 71.5 ± 0.2 | 72.1 ± 0.1 | 72.1 ± 0.2 |
| | 8192 | 71.6 ± 0.1 | 72.2 ± 0.1 | 72.1 ± 0.3 |

| Dataset | Batch Size | MVS-GCN | +REST | +REST-IS |
|---|---|---|---|---|
| **Products** | 10000 | 76.7 ± 0.2 | 77.9 ± 0.3 | 78.2 ± 0.3 |
| | 50000 | 76.9 ± 0.1 | 78.1 ± 0.2 | 78.3 ± 0.3 |
| **Reddit** | 256 | 93.8 ± 0.2 | 94.5 ± 0.2 | 94.7 ± 0.1 |
| | 512 | 94.2 ± 0.1 | 94.8 ± 0.1 | 95.0 ± 0.2 |
| | 2048 | 94.9 ± 0.1 | 95.2 ± 0.1 | 95.2 ± 0.2 |
| **Arxiv** | 128 | 70.9 ± 0.2 | 71.9 ± 0.2 | 72.0 ± 0.3 |
| | 512 | 71.4 ± 0.1 | 72.4 ± 0.1 | 72.5 ± 0.3 |
| | 2048 | 71.5 ± 0.1 | 72.0 ± 0.1 | 72.1 ± 0.1 |
| | 8192 | 71.6 ± 0.1 | 72.1 ± 0.1 | 72.0 ± 0.2 |

• Our algorithms outperform all the baselines by a **substantial margin, particularly in scenarios with significant staleness**, such as large datasets or small batch sizes. We also show similar results of GraphFM in Appendix J. For example, on the ogbn-arxiv dataset, we observe a performance boost of 2.4% with REST and 3.0% with REST-IS using a batch size of 5 and APPNP as backbone. Similarly, on ogbn-products, our algorithms outperform GAS by 4.7% with REST and 5.4% with REST-IS under the same batch size setting. The similar phenomenon is noticeable when our approach is implemented on VR-GCN and MVS-GCN.

• Our proposed algorithms exhibit exceptional adaptabil-

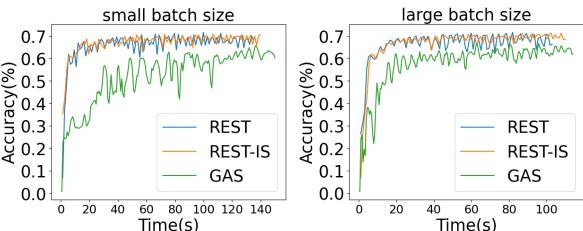

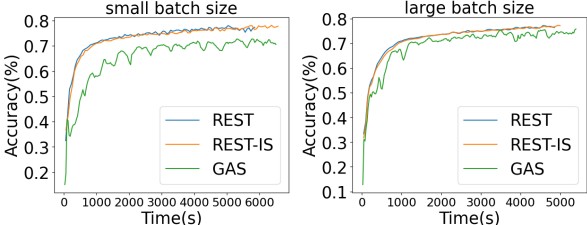

Figure 6: Convergence on ogbn-arxiv.

Figure 7: Convergence on ogbn-products.

ity by seamlessly integrating with all historical embedding methods and surpassing the performance of three prominent historical embedding methods. This showcases the flexibility and universality of our algorithms.

• It is important to note that our proposed methods are designed to **alleviate the staleness issues present in those historical embedding-based baselines**. Staleness only becomes significant when dealing with large-scale datasets, which explains why our model demonstrates substantially larger improvements on datasets such as ogbn-products, ogbn-papers100M, and MAG240M compared to other datasets (see Appendix E). These results clearly indicate that our algorithm effectively mitigates staleness, and substantial enhancements can be easily achieved with different GNN backbones.

## 4.3. Efficiency Analysis

To verify the efficiency of our proposed approach, we provide empirical efficiency analysis compared with one scalable GNNs, GraphSAGE and two historical embedding methods, VR-GCN and GAS in Table 4 and Table 5. Specifically, we measure the memory usage, **total** running time and the approximate number of epochs until convergence for training process on the ogbn-arxiv and ogbn-products under different batch sizes. Note that, the number under "Batch Size" indicates the number of sampled clusters in one batch in Table 4. We used a server, and all experiments were run on a single NVIDIA A6000 GPU. To ensure a fair comparison, we employ the same official implementations for all baseline methods (Fey et al., 2021; Shi et al., 2023) as well as our proposed method. For GAS, we use the APPNP with 5 aggregation layers as the GNN backbone and keep all other hyperparameters the same to make a fair comparison. We also include the performance on the same table for the convenience.

It's important to note that GraphSAGE may still encounter the neighbor explosion problem, leading to out-of-memory (OOM) errors for ogbn-products and significantly higher memory costs for ogbn-arxiv in our experiments. REST demonstrates a comparable running time to GraphSAGE while achieving superior performance and significantly lower memory costs. Compared to GAS and VR-GCN, our approach is designed to be compatible with any histori-

cal embedding model, addressing the staleness issue without modifying the model architecture. As a result, it preserves the memory efficiency of the original baselines while achieving better performance and expediting the training process. This is attributed to REST's faster convergence (see Theorem 3.2), which requires substantially fewer epochs to converge, despite a longer per-epoch runtime due to multiple forward passes (see Figure 8, 9, 10 and 11 in Appendices).

## 4.4. Convergence Analysis

We present a comparison of convergence analysis by showing test accuracy with respect to the running time for both GAS and our algorithms on the ogbn-arxiv and ogbn-products datasets with different batch sizes. Specifically, we represent small and large batch sizes using 5 and 10 clusters for both datasets. The convergence curves are shown in Figures 6 and 7, respectively. The results illustrate that our model not only achieves superior performance but also converges faster. This advantage is more evident when there is significant staleness, such as when smaller batch sizes are used or with larger datasets as shown in the first subfigures in Figures 6 and 7. Importantly, the convergence of GAS is significantly influenced by staleness. However, our model's convergence speed only experiences a slight decrease as staleness becomes larger, which supports the convergence advantage discussed in introduction. Compared to REST, REST-IS appears to be more stable after convergence. We conclude that this is attributed to that we continually refresh the embeddings of important nodes. All observations indicate that our algorithm possesses advantages in convergence beyond mere performance metrics. We also conducted similar experiments demonstrating convergence over epochs using GAS and VR-GCN in Appendix C. We can draw the same conclusion that convergence is accurately achieved by our algorithm, regardless of the historical embedding methods used. Please also refer to Appendix K for SAGE.

## 4.5. Ablation Study

In this subsection, we present detailed ablation studies on the impact of important hyperparameters and the generality of our approach.

**Forward batch size $B$ for memory table update.** In addition to the flexible adjustment of the updating frequency

Table 4: Memory usage (MB), Total running time (seconds) and Epochs on ogbn-arxiv and ogbn-products.

| Dataset | Parts | Batch Size | MEMORY(MB) | | | | TOTAL TIME(S) / # EPOCHS | | | | ACCURACY | | | |
|---|---|---|---|---|---|---|---|---|---|---|---|---|---|---|
| | | | SAGE | GAS | REST | REST-IS | SAGE | GAS | REST | REST-IS | SAGE | GAS | REST | REST-IS |
| ogbn-products | 40 | 5 | OOM | 8913 | 9295 | 10059 | N/A | 2600 / 75 | 1170 / 20 | 1312 / 18 | N/A | 75.2 | 79.7 | **80.5** |
| | | 10 | OOM | 13406 | 13495 | 14753 | N/A | 1890 / 65 | 940 / 18 | 1200 / 18 | N/A | 76.9 | 80.0 | **80.4** |
| ogbn-arxiv | 40 | 5 | 3011 | 790 | 837 | 703 | 39.3 / 35 | 67.5 / 60 | 39 / 16 | 42 / 15 | 70.9 | 69.4 | 71.7 | **72.1** |
| | | 10 | 3156 | 997 | 1115 | 948 | 37.2 / 33 | 54.0 / 58 | 31.5 / 18 | 33 / 18 | 71.2 | 70.1 | 72.3 | **72.4** |
| | | 20 | 3323 | 1486 | 1505 | 1262 | 25.8 / 32 | 39.6 / 50 | 24.0 / 17 | 19.5 / 17 | 71.5 | 71.5 | 72.4 | **72.4** |

Table 5: Memory usage (MB), Total running time (seconds) and Epochs on ogbn-products.

| Batch Size | MEMORY(MB) | | | TOTAL TIME(S)/ # EPOCHS | | | ACCURACY(%) | | |
|---|---|---|---|---|---|---|---|---|---|
| | VR-GCN | REST | REST-IS | VR-GCN | REST | REST-IS | VR-GCN | REST | REST-IS |
| 10000 | 8730 | 8733 | 9119 | 2892 / 25 | 2616 / 12 | 2275 / 11 | 76.3 | 77.4 | 78.0 |
| 50000 | 15402 | 17585 | 16322 | 1104 / 21 | 936 / 11 | 920 / 10 | 76.4 | 77.5 | 78.1 |

$f$, our algorithm offers another advantage: the batch size during the memory table updating process can also be reasonably increased since it does not require significant memory for backward propagation. Please refer Appendix F for all results. The presented results indicate that a larger batch size used in updating the memory table has the potential to further enhance performance by reducing staleness.

**Forward computation frequency $F$.** We also evaluate the performance under different frequencies on the ogbn-products dataset in Appendix D. We observe that the performance gradually increases with the updating frequency and higher frequencies tend to enhance convergence, validating the conclusion that staleness is a key factor affecting performance and our algorithm demonstrates effectiveness in addressing this issue.

**Bidirectional REST.** While existing historical embedding methods like GAS and GraphFM consider all available neighborhood information, they fail to aggregate gradient information from these neighbors because they reside in memory and are excluded from computation graphs. One straightforward solution is to maintain an additional memory per layer to store historical gradients, similar to the forward pass in GAS. However, backward propagation encounters the same bottleneck of staleness. We can leverage the idea of REST to increase the frequency of gradient memory updates, thereby reducing gradient staleness bidirectionally.

Specifically, we select a frequency $\tilde{F}$ where $\tilde{F} \leq F$, indicating that we can easily choose $\tilde{F}$ out of $F$ batches for gradient computation during the execution of line 7 in Algorithm 1, updating the historical gradient memory without altering model parameters. Analogous to the forward pass, we conduct backward propagation $\tilde{F}$ times to refresh the gradient memory bank without updating any model parameters. Subsequently, we execute a standard backward propagation to update the model parameters (line 10 in Algorithm 1). Because LMC is the only baseline that incorporates historical gradients via momentum, we include LMC as our only baseline and report its results in Table 6. We set $\tilde{F}=1$ and use GCN as backbone for simplicity. All LMC experiments follow the settings provided in the authors' official repository. From the results, we observe that bidirectional REST improves accuracy and accelerates training while keeping memory cost comparable. In contrast, LMC performs poorly because it cannot fundamentally resolve the gradient staleness issue. Overall, these findings underscore the versatility and benefits of our approach.

Table 6: Performance (%), Memory usage(MB) and Total running time(seconds) of Bidirectional REST

| Models | OGBN-ARXIV | | | OGBN-PRODUCTS | | |
|---|---|---|---|---|---|---|
| | ACC | MEMORY | RUNNING TIME | ACC | MEMORY | RUNNING TIME |
| LMC | 71.4 | 558 | 66 | 77.5 | 10982 | 1520 |
| LMC+REST | **72.6** | 584 | 41 | **80.1** | 11139 | 925 |

**Other Studies.** We present the ablation study as follows in Appendices: (1) Comparison with more baselines in Appendices G and O; (2) Performance and efficiency on more large scale datasets in Appendix E ; (2) Memory persistence and Embedding approximation errors of REST in Appendix H; (3) Various aggregation layers in Appendix I; (4) Further analyses with various baselines in Appendices J and K. (5) Difference between REST and other techniques in Appendices L, N and Q. (6) Broad impact and generalizability analysis of REST in in Appendix P.

## 5. Conclusion

In this paper, we first conduct a highly comprehensive study of existing historical embedding methods and conclude that the primary reason for their inferior performance and slow convergence originates from staleness. Instead of merely attempting to control staleness, we aim to address this issue at its source. Through both theoretical and empirical analyses, we identify the root cause as the disparity in the frequency of updating the memory table and model parameters. Then, we propose a simple yet highly effective algorithm by decoupling forward and backward propagations and executing them at different frequencies. Comprehensive experiments demonstrate its superior ability to resolve the staleness issue, leading to significantly improved prediction performance, faster convergence, and greater flexibility on large-scale graph datasets.

## Impact Statement

Graph Neural Networks (GNNs) have become essential for learning on graph-structured data, yet scalability remains a significant challenge due to memory limitations and computational overhead. Historical embedding methods have been widely adopted to improve scalability but suffer from substantial bias caused by stale feature history. This paper first provides the most comprehensive analysis, combining theoretical and empirical studies to uncover the root causes of staleness. To address this issue, we introduce REST, a simple yet highly effective approach that reduces feature staleness in historical embedding methods. By decoupling forward and backward propagations and dynamically adjusting their execution frequencies, REST effectively mitigates the impact of stale embeddings, resulting in significantly enhanced prediction accuracy and faster convergence across large-scale graph datasets.

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

## A. Proof of Theorem 1

**Theorem A.1** (Approximation Error)**.** *Consider a L-layers GNN $f_\theta^{(l)}(h)$ with Lipschitz constant $\alpha^{(l)}$, UPDATE$_\theta^{(l)}$ function with Lipschitz constant $\beta$, $l = 1, \ldots, L$. $\nabla L_\theta$ has Lipschitz constant $\varepsilon$. If $\forall v \in V$, $||\bar{h}_v^{(l)} - h_v^{(l)}||$ denotes the staleness between the historical embeddings and the true embeddings from full batch aggregations, $\mathcal{N}(v)$ is the neighborhood set of $v$, then the approximation error of gradients is bounded by:*

$$||\nabla L_\theta(\tilde{h}_v^{(L)}) - \nabla L_\theta(h_v^{(L)})|| \leq \varepsilon \sum_{l=1}^{L} \left( \prod_{k=l+1}^{L} \alpha^{(k)} \beta |\mathcal{N}(v)| * ||\tilde{L}_{v,}|| * ||\bar{h}^{(k-1)} - h^{(k-1)}|| \right).$$

### A.1. Detailed Proof:

*Proof.* Suppose $\tilde{f}_\theta^{(l)}$ is a historical embedding-based GNN with L-layers, then the whole GNN model can be defined as $\tilde{h}^{(L)} = \tilde{f}_\theta^{(L)} \circ \tilde{f}_\theta^{(L-1)} \circ \cdots \circ \tilde{f}_\theta^{(1)}$, similarly, the full batch GNN can be defined as: $h^{(L)} = f_\theta^{(L)} \circ f_\theta^{(L-1)} \circ \cdots \circ f_\theta^{(1)}$, then:

$$||\tilde{h}^{(L)} - h^{(L)}|| = ||\tilde{f}_\theta^{(L)} \circ \tilde{f}_\theta^{(L-1)} \circ \cdots \circ \tilde{f}_\theta^{(1)} - f_\theta^{(L)} \circ f_\theta^{(L-1)} \circ \cdots \circ f_\theta^{(1)}|| \tag{10}$$

$$= ||\tilde{f}_\theta^{(L)} \circ \tilde{f}_\theta^{(L-1)} \circ \cdots \circ \tilde{f}_\theta^{(1)} - \tilde{f}_\theta^{(L)} \circ \tilde{f}_\theta^{(L-1)} \circ \cdots \circ f_\theta^{(1)} \tag{11}$$

$$+ \tilde{f}_\theta^{(L)} \circ \tilde{f}_\theta^{(L-1)} \circ \cdots \circ \tilde{f}_\theta^{(2)} \circ f_\theta^{(1)} - \tilde{f}_\theta^{(L)} \circ \tilde{f}_\theta^{(L-1)} \circ \cdots \circ f_\theta^{(2)} \circ f_\theta^{(1)} - \ldots \tag{12}$$

$$+ \tilde{f}_\theta^{(L)} \circ f_\theta^{(L-1)} \circ \cdots \circ f_\theta^{(1)} - f_\theta^{(L)} \circ f_\theta^{(L-1)} \circ \cdots \circ f_\theta^{(1)}|| \tag{13}$$

$$\leq ||\tilde{f}_\theta^{(L)} \circ \tilde{f}_\theta^{(L-1)} \circ \cdots \circ \tilde{f}_\theta^{(1)} - \tilde{f}_\theta^{(L)} \circ \tilde{f}_\theta^{(L-1)} \circ \cdots \circ f_\theta^{(1)}|| + \tag{14}$$

$$\cdots + ||\tilde{f}_\theta^{(L)} \circ f_\theta^{(L-1)} \circ \cdots \circ f_\theta^{(1)} - f_\theta^{(L)} \circ f_\theta^{(L-1)} \circ \cdots \circ f_\theta^{(1)}|| \tag{15}$$

$$= \sum_{k=1}^{L} \left( \prod_{l=k+1}^{L} \alpha^{(l)} ||\tilde{f}_\theta^{(k)} \circ f_\theta^{(k-1)} \circ \cdots \circ f_\theta^{(1)} - f_\theta^{(k)} \circ f_\theta^{(k-1)} \circ \cdots \circ f_\theta^{(1)}|| \right) \tag{16}$$

According to the rule of message passing, assuming update function, aggregation function and message passing function at $k$-th layer is denoted by $g_{update}^{(k)}$, $g_{agg}^{(k)}$ and $g_{msp}^{(k)}$, respectively, then:

$$||\tilde{f}_\theta^{(k)} \circ f_\theta^{(k-1)} \circ \cdots \circ f_\theta^{(1)} - f_\theta^{(k)} \circ f_\theta^{(k-1)} \circ \cdots \circ f_\theta^{(1)}|| \tag{17}$$

$$= ||g_{update}^{(k)} \left( h_v^{(k-1)}, g_{agg}^{(k)} \left( g_{msp}^{(k)}(\bar{h}^{(k-1)}) \right) \right) - g_{update}^{(k)} \left( h_v^{(k-1)}, g_{agg}^{(k)} \left( g_{msp}^{(k)}(h^{(k-1)}) \right) \right) || \tag{18}$$

$$\leq \beta * || \sum_{\mathcal{N}(v)} \tilde{L}_v, \bar{h}^{(k-1)} - \sum_{\mathcal{N}(v)} \tilde{L}_v, h^{(k-1)}|| \tag{19}$$

$$\leq \beta |\mathcal{N}(v)| * ||\tilde{L}_v, \bar{h}^{(k-1)} - \tilde{L}_v, h^{(k-1)}|| \tag{20}$$

$$\leq \beta |\mathcal{N}(v)| * ||\tilde{L}_v,|| * ||\bar{h}^{(k-1)} - h^{(k-1)}|| \tag{21}$$

From $||\nabla L_\theta(\tilde{h}_v^{(L)}) - \nabla L_\theta(h_v^{(L)})|| \leq \varepsilon ||\tilde{h}_v^{(L)} - h_v^{(L)}||$, we can get final conclusion by combining all previous steps together:

$$||\nabla L_\theta(\tilde{h}_v^{(L)}) - \nabla L_\theta(h_v^{(L)})|| \leq \varepsilon \sum_{l=1}^{L} \left( \prod_{k=l+1}^{L} \alpha^{(k)} \beta |\mathcal{N}(v)| * ||\tilde{L}_v,|| * ||\bar{h}^{(l-1)} - h^{(l-1)}|| \right) \tag{22}$$

$\square$

### A.2. Lipschitz Continuity Assumption:

Consider a GNN with $L$ layers, where the embedding at layer $\ell$ is given by

$$h^{(\ell)} = f_\theta^{(\ell)}(h^{(\ell-1)}), \tag{23}$$

with $h^{(0)}$ as the input features. A typical layer can be expressed as

$$h_v^{(\ell+1)} = \text{UPDATE}\Big(h_v^{(\ell)},\ \text{AGG}\big(\{h_u^{(\ell)} : u \in \mathcal{N}(v)\}\big)\Big). \tag{24}$$

Then, the following assumptions usually hold in the literature (e.g., in GAS (Fey et al., 2021)):

- The aggregation function evaluated on the embeddings, AGG, is $\alpha$-Lipschitz:

$$\|\text{AGG}(X) - \text{AGG}(Y)\| \le \alpha \|X - Y\|, \tag{25}$$

  where functions such as mean, sum, and max are all Lipschitz continuous.

- The update function evaluated on the embeddings, UPDATE, is $\beta$-Lipschitz:

$$\|\text{UPDATE}(X) - \text{UPDATE}(Y)\| \le \beta(X - Y). \tag{26}$$

Furthermore, commom activation functions and loss functions with respect to the predictions are also Lipschitz continuous. When these components are composed to form the full GNN, the overall Lipschitz continuity is preserved, with the constant being a function of the individual layer constants.

## B. Proof of Theorem 2

**Theorem B.1.** *Given the upper bound of the expectation of gradients' norm in the state-of-the-art historical embeddings methods such as GAS, LMC, and REST, which is*

$$E[||\nabla_\theta L(\theta_R)||_2] \le \left( \frac{2(L(\theta_1) - L_\theta^* + G_\theta)}{N^{\frac{1}{3}}} + \frac{\epsilon G_\theta}{N^{\frac{2}{3}}} + \frac{G_\theta}{N(1 - \sqrt{\rho})} \right)^{\frac{1}{2}} \tag{27}$$

*where $G_\theta$ is the upper bound of gradient approximation error, $N$ is the number of iterations, $R$ is chosen uniformly from $[N]$, $\rho \in (0, 1)$. Based on Theorem 3.1, the upper bound $G_\theta$ of REST is tighter than that of existing historical embedding methods. Consequently, the convergence speed of REST surpasses that of existing works.*

*Proof.* Suppose that function $L : \mathbb{R}^n \to \mathbb{R}$ is continuously differentiable. Consider an optimization algorithm with any bounded initialization $x_1$ and an update rule in the form of

$$x_{k+1} = x_k - \eta\, d(x_k), \tag{28}$$

where $\eta > 0$ is the learning rate and $d(x_k)$ is the estimated gradient that can be viewed as a stochastic vector depending on $x_k$. Let the estimation error of the gradient be

$$\Delta_k = d(x_k) - \nabla L(x_k). \tag{29}$$

Suppose that (Chen et al., 2017; Shi et al., 2023):

1. The optimal value $L^* = \inf_x L(x)$ is bounded;

2. The gradient of $L$ is $\epsilon$-Lipschitz, i.e.,

$$\|\nabla L(y) - \nabla L(x)\|_2 \le \epsilon \| y - x \|_2, \quad \forall x, y \in \mathbb{R}^n; \tag{30}$$

3. There exists $G_\theta > 0$ (independent of $\eta$) such that [1]

$$\mathbb{E}\big[\|\Delta_k\|_2^2\big] \le G_\theta, \quad \forall\, k \in \mathbb{N}^*; \tag{31}$$

4. There exist $N \in \mathbb{N}^*$ and $\rho \in (0, 1)$ (both independent of $\eta$) such that

$$\big| \mathbb{E}\langle \nabla L(x_k),\, \Delta_k \rangle \big| \le G_\theta\big(\eta^{1/2} + \rho_{\frac{k-1}{2}} + \tfrac{1}{N^{1/3}}\big), \quad \forall\, k \in \mathbb{N}^*, \tag{32}$$

  where $G_\theta$ is the same constant as that in Condition 3. [1]

Then by letting

$$\eta \;=\; \min\Big\{ \tfrac{1}{\epsilon}, \; \tfrac{1}{N^{2/3}} \Big\}, \tag{33}$$

we have

$$\mathbb{E}\big[\, \|\nabla L(x_R)\|_2^2 \big] \;\leq\; \frac{2\big(L(x_1) \,-\, L^* \,+\, G_\theta\big)}{N^{1/3}} \;+\; \frac{\epsilon\, G_\theta}{N^{2/3}} \;+\; \frac{G_\theta}{N\big(1 - \sqrt{\rho}\big)}, \tag{34}$$

where $R$ is chosen uniformly at random from $\{1, \ldots, N\}$.

As the gradient of $L$ is $\epsilon$-Lipschitz, we have

$$L(y) \;=\; L(x) \;+\; \int_x^y \nabla L(z)\, dz \;=\; L(x) \;+\; \int_0^1 \Big\langle \nabla L\big(x + t(y-x)\big),\, y - x \Big\rangle dt. \tag{35}$$

Hence,

$$L(y) = L(x) \;+\; \Big\langle \nabla L(x),\, y - x \Big\rangle \;+\; \int_0^1 \Big\langle \nabla L\big(x + t(y-x)\big) \,-\, \nabla L(x),\, y - x \Big\rangle dt \tag{36}$$

$$\leq \; L(x) \;+\; \langle \nabla L(x),\, y - x \rangle \;+\; \int_0^1 \Big\| \nabla L\big(x + t(y-x)\big) \,-\, \nabla L(x) \Big\|_2 \, \| y - x \|_2 \, dt \tag{37}$$

$$\leq \; L(x) \;+\; \langle \nabla L(x),\, y - x \rangle \;+\; \int_0^1 \epsilon\, t \, \| y - x \|_2^2 \, dt \tag{38}$$

$$\leq \; L(x) \;+\; \langle \nabla L(x),\, y - x \rangle \;+\; \frac{\epsilon}{2} \, \| y - x \|_2^2. \tag{39}$$

Now let $x_{k+1} = x_k - \eta\, d(x_k)$ and $\Delta_k = d(x_k) \,-\, \nabla L(x_k)$. Then

$$L\big(x_{k+1}\big) \;\leq\; L\big(x_k\big) \;-\; \eta \Big\langle \nabla L(x_k),\, d(x_k) \Big\rangle \;+\; \frac{\epsilon\, \eta^2}{2} \, \big\| d(x_k) \big\|_2^2. \tag{40}$$

Taking the total expectation on both sides and rearranging terms yields

$$\mathbb{E}\big[L\big(x_{k+1}\big)\big] \;\leq\; \mathbb{E}\big[L\big(x_k\big)\big] \;-\; \eta\, \mathbb{E}\Big\langle \nabla L(x_k),\, \nabla L(x_k) + \Delta_k \Big\rangle \;+\; \frac{\epsilon\, \eta^2}{2} \, \mathbb{E}\big[\|d(x_k)\|_2^2\big] \tag{41}$$

$$= \; \mathbb{E}\big[L\big(x_k\big)\big] \;-\; \eta\, \mathbb{E}\big[\|\nabla L(x_k)\|_2^2\big] \;-\; \eta\, \mathbb{E}\Big[\langle \nabla L(x_k),\, \Delta_k \rangle\Big] \;+\; \frac{\epsilon\, \eta^2}{2} \, \mathbb{E}\big[\|\nabla L(x_k) + \Delta_k\|_2^2\big]. \tag{42}$$

Using $\|\nabla L(x_k) + \Delta_k\|_2^2 \leq 2\|\nabla L(x_k)\|_2^2 + 2\|\Delta_k\|_2^2$ and the assumption $\mathbb{E}\big[\|\Delta_k\|_2^2\big] \leq G_\theta$, we obtain

$$\mathbb{E}\big[L(x_{k+1})\big] \;\leq\; \mathbb{E}\big[L\big(x_k\big)\big] \;-\; \eta\, \mathbb{E}\big[\|\nabla L(x_k)\|_2^2\big] \;-\; \eta\, \mathbb{E}\Big[\langle \nabla L(x_k),\, \Delta_k \rangle\Big] \;+\; \epsilon\, \eta^2\, \mathbb{E}\big[\|\nabla L(x_k)\|_2^2\big] \;+\; \epsilon\, \eta^2\, \mathbb{E}\big[\|\Delta_k\|_2^2\big] \tag{43}$$

$$\leq \; \mathbb{E}\big[L\big(x_k\big)\big] \;-\; \eta\big(1 - \epsilon\, \eta\big)\, \mathbb{E}\big[\|\nabla L(x_k)\|_2^2\big] \;-\; \eta\, \mathbb{E}\big[\langle \nabla L(x_k),\, \Delta_k \rangle\big] \;+\; \epsilon\, \eta^2\, G_\theta. \tag{44}$$

Summing over $k = 1, \ldots, N$, dividing by $N$, and rearranging imply

$$\frac{1}{N} \sum_{k=1}^N \mathbb{E}\big[\|\nabla L(x_k)\|_2^2\big] \;\leq\; \frac{L(x_1) \,-\, \mathbb{E}\big[L(x_{N+1})\big]}{\eta\big(1 - \epsilon\, \eta\big) N} \;-\; \frac{\mathbb{E}\Big[\sum_{k=1}^N \langle \nabla L(x_k),\, \Delta_k \rangle\Big]}{(1 - \epsilon\, \eta) N} \;+\; \frac{\epsilon\, \eta\, G_\theta}{1 - \epsilon\, \eta}. \tag{45}$$

Since $L^* = \inf_x L(x)$ is bounded and $x_1$ is bounded, $L(x_1) - \mathbb{E}[L(x_{N+1})] \leq L(x_1) - L^*$. Also note that $\big|\mathbb{E}\langle \nabla L(x_k),\, \Delta_k \rangle\big| \leq G_\theta\big(\eta^{1/2} + \rho^{(k-1)/2} + N^{-1/3}\big)$ by assumption. Hence,

$$\frac{1}{N} \sum_{k=1}^{N} \mathbb{E}\big[\|\nabla L(x_k)\|_2^2\big] \leq \frac{L(x_1) - L^*}{N \eta (1 - \epsilon \eta)} + \frac{\epsilon \eta G_\theta}{1 - \epsilon \eta} + \frac{1}{N(1 - \epsilon \eta)} \sum_{k=1}^{N} \Big| \mathbb{E}\langle \nabla L(x_k), \Delta_k \rangle \Big| \tag{46}$$

$$\leq \frac{L(x_1) - L^*}{N \eta (1 - \epsilon \eta)} + \frac{\epsilon \eta G_\theta}{1 - \epsilon \eta} + \frac{G_\theta}{(1 - \epsilon \eta) N} \sum_{k=1}^{N} \Big( \eta^{1/2} + \rho^{(k-1)/2} + N^{-1/3} \Big). \tag{47}$$

Since $\sum_{k=1}^{N} \rho^{\frac{k-1}{2}} \leq \frac{1}{1 - \sqrt{\rho}}$, we then get

$$\frac{1}{N} \sum_{k=1}^{N} \mathbb{E}\big[\|\nabla L(x_k)\|_2^2\big] \leq \frac{L(x_1) - L^*}{N \eta (1 - \epsilon \eta)} + \frac{\epsilon \eta G_\theta}{1 - \epsilon \eta} + \frac{G_\theta}{N(1 - \epsilon \eta)} \Big( N \eta^{1/2} + \frac{1}{1 - \sqrt{\rho}} + N^{2/3} \Big). \tag{48}$$

Note that

$$\mathbb{E}\big[\|\nabla L(x_R)\|_2^2\big] = \mathbb{E}_R\big[\|\nabla L(x_R)\|_2^2 \,\big|\, R\big] = \frac{1}{N} \sum_{k=1}^{N} \mathbb{E}\big[\|\nabla L(x_k)\|_2^2\big] \tag{49}$$

Finally choose $\eta = \min\{\frac{1}{\epsilon}, \frac{1}{N^{2/3}}\}$, it follows that

$$\mathbb{E}\big[\|\nabla L(x_R)\|_2^2\big] = \frac{1}{N} \sum_{k=1}^{N} \mathbb{E}\big[\|\nabla L(x_k)\|_2^2\big] \leq \frac{2\left(L(x_1) - L^* + G_\theta\right)}{N^{1/3}} + \frac{\epsilon G_\theta}{N^{2/3}} + \frac{G_\theta}{N(1 - \sqrt{\rho})}, \tag{50}$$

which completes the proof.

$\square$

## C. Convergence

We replicated the convergence analysis outlined in our main paper (Figure 6 and Figure 7), using VR-GCN and GAS, but with epochs as the x-axis. The results are shown in Figure 8 and 9 for GAS and Figure 10 and 11 for VR-GCN. GAS settings remained consistent with those specified in the main paper. For VR-GCN, we experimented with batch sizes of 128 and 2048 for the ogbn-arxiv dataset, and 10000 and 50000 for ogbn-products. For GAS, our findings align with those of the main paper: our algorithm not only achieves superior performance but also converges more rapidly with much less epochs. For VR-GCN, when dealing with small datasets with large batch sizes, VR-GCN exhibits comparable convergence to our approach, as staleness isn't a significant factor. However, as the dataset size increases or the batch size decreases, VR-GCN's convergence deteriorates significantly, whereas our algorithm maintains robust convergence. Furthermore, after convergence, REST-IS demonstrates greater resilience than REST in scenarios where staleness plays a significant role.

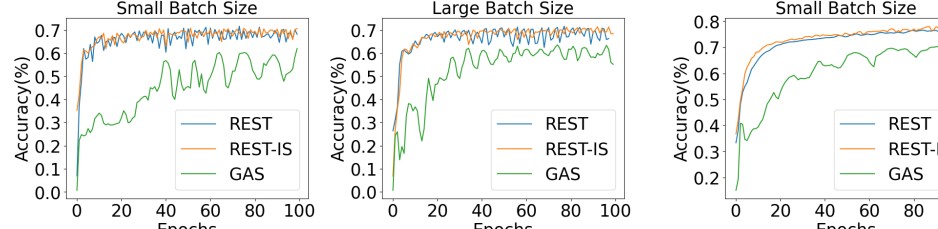

Figure 8: ogbn-arxiv

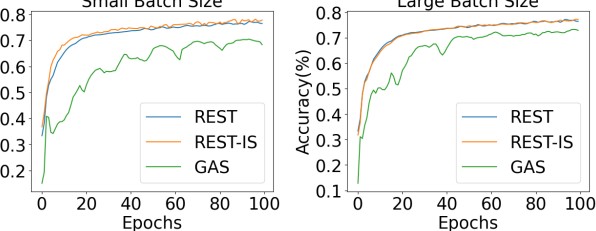

Figure 9: ogbn-products

## D. Forward computation frequency $F$

We evaluate the performance under different frequencies on the ogbn-products dataset. We select the case with 5 clusters (40 clusters in total) for APPNP and report the performance in Figure 12. We observe that the performance gradually increases

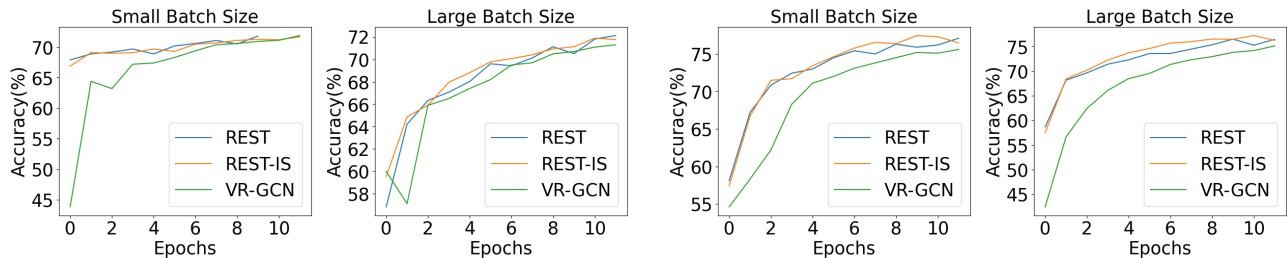

Figure 10: ogbn-arxiv                    Figure 11: ogbn-products

with the updating frequency, validating the conclusion that staleness is a key factor affecting performance and our algorithm demonstrates effectiveness in addressing this issue.

Furthermore, we also present the efficiency analysis with different frequency F in Table 7, accompanied by convergence curves corresponding to different frequencies in Figure 13 and 14. As observed from the results, higher frequencies tend to enhance convergence, as shown in Figure 13 (epoch as unit). However, they also entail extra computation overhead. Hence, the actual convergence time remains relatively consistent across different frequencies, as illustrated in Figure 14 (time as unit). Nevertheless, all cases exhibit significant improvements compared to GAS.

Table 7: Memory usage (MB) and running time (seconds) with different frequency.

| Dataset | Freduency | MEMORY(MB) | TIME(S) |
|---------|-----------|------------|---------|
| ogbn-products | 2 | 9295 | 1053 |
| | 3 | 9295 | 1188 |
| | 4 | 9295 | 1200 |
| | 5 | 9295 | 1204 |

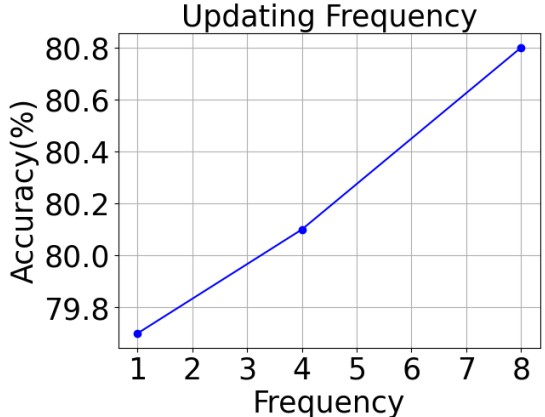

Figure 12: Different Frequency F

# E. Performance and Efficiency on Larger Datasets

In this section, we demonstrate the effectiveness on larger datasets, which typically exhibit more severe staleness problems. Given our limited computational resources, we chose to add experiments on ogbn-papers100M and MAG240M, which are significantly larger in scale compared to other commonly used datasets and are highly representative. Note that for MAG240M, we follow the common practice of homogenizing it (Hu et al., 2021; Jiang et al., 2023). We report the performance and efficiency in Table 8.

The larger size of the ogbn-papers100M and MAG240M dataset exacerbates the staleness issue for GAS, leading to

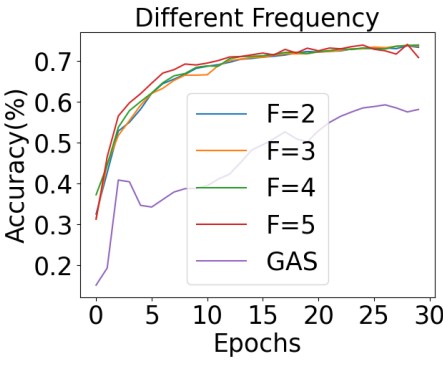

Figure 13: Convergence w.r.t epochs

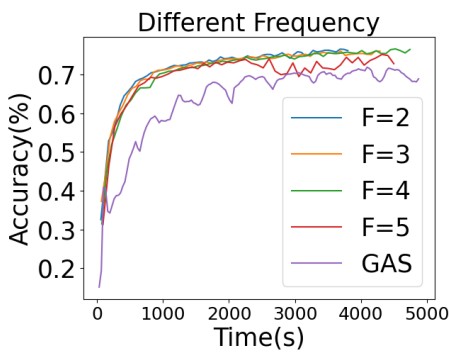

Figure 14: Convergence w.r.t time

decreased accuracy and slower convergence, as anticipated. In contrast, REST demonstrates significantly improved accuracy and efficiency, along with accelerated convergence. This conclusion aligns with our main claim in the main paper that REST has a strong ability to address the staleness issue.

Table 8: Memory usage (MB) and running time (seconds) on larger datasets.

| Datasets | Models | Accuracy | MEMORY(MB) | TIME(S) |
|---|---|---|---|---|
| ogbn-papers100M | GAS | 64.9 | 15705 | 8840 |
| | GAS + REST | 67.6 | 16808 | 4100 |
| MAG240M | GAS | 61.2 | 31256 | 42603 |
| | GAS + REST | 67.6 | 32062 | 25355 |

## F. Forward batch size $B$ for memory table update.

In addition to the flexible adjustment of the updating frequency $F$, our algorithm offers another advantage: the batch size during the memory table updating process (line 7 in Algorithm 1) can also be reasonably increased since it does not require significant memory for backward propagation. We explore the effect of varying batch size on the final performance using GCN, APPNP, and GCNII as the backbone. We also include GAS for comparison. Specifically, we illustrate three scenarios for the batch size: (1) *Same*: identical to the batch size used for updating the model (consistent with the setting in Table 2); (2) *Half*: half batch (encompassing half of all clusters); and (3) *Full*: full batch (encompassing all clusters (whole graph)). We use frequency $F$ =1 for all cases. We show the results of three cases in Figure 15.

The presented results indicate that a larger batch size used in updating the memory table has the potential to further enhance performance by reducing staleness. This is because a large batch size accelerates the memory table updates, which is equivalent to using a higher frequency $F$. However, the improvement is marginal and comes with additional memory costs. Consequently, we opt to maintain the same batch size to keep the memory efficiency of our proposed algorithm.

## G. More Recent Baselines

Several historical embedding methods have been proposed recently, including LMC (Shi et al., 2023), which incorporates a memory table for neighbors' gradients, and Refresh (Huang et al., 2023), which utilizes staleness scores and gradient changes as metrics to selectively update the memory table. However, as discussed in related works in Appendix M, these methods still struggle to address the staleness issue at its source, and their techniques are orthogonal to ours. Therefore, to provide a comprehensive analysis, we have included a performance comparison between REST, Refresh, and LMC in Table 9. We employ GCN as the GNN backbone model and include the GAS case in the table for convenience. From the results, we observe that REST can be easily combined with existing works and significantly outperforms them. For example, it achieves a 2.6% increase over LMC on the ogbn-products dataset.

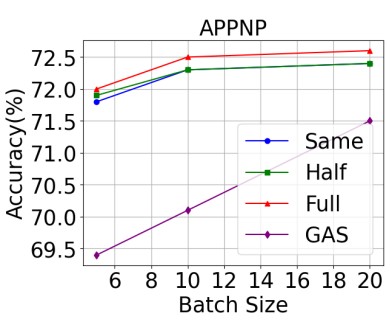
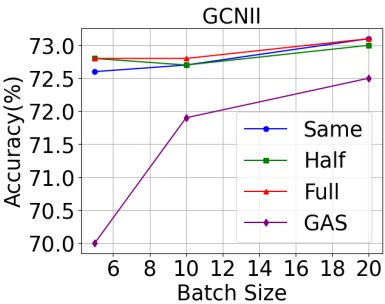

Figure 15: The impact of batch size (a) GCN, (b) APPNP, and (c) GCNII

Table 9: Performance Comparison on ogbn-arxiv and ogbn-products.

| Models | OGBN-ARXIV | OGBN-PRODUCTS |
|---|---|---|
| Refresh | 70.5 | 78.3 |
| GAS | 71.7 | 76.7 |
| GAS+REST | 72.2 | 79.6 |
| LMC | 71.4 | 77.5 |
| LMC+REST | 72.6 | 80.1 |

## H. Memory persistence and embedding approximation errors of REST

We present two additional results on memory persistence and approximation error comparison, similar to those in our preliminary study, between GAS and REST+GAS on the ogbn-arxiv dataset, as follows:

(1) Memory Persistence (Figure 16): It is evident that persistence continuously decreases with the increase in updating frequency, regardless of the batch size used. This directly demonstrates that staleness is reduced by REST.

(2) Approximation Error (Figure 17): The decrease in error value is also noticeable when compared to the case where REST is utilized. This provides another straightforward evidence showing the effectiveness of REST.

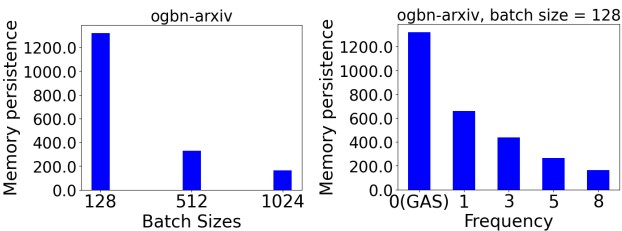

Figure 16: Left: GAS, Right: REST + GAS.

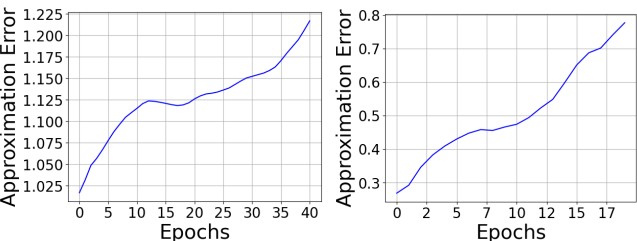

Figure 17: Left: GAS, Right: REST + GAS

## I. Various aggregation layers

We introduce an additional experiment conducted on the ogbn-arxiv dataset to evaluate the influence of the number of aggregation layers on performance. For simplicity, we opt to employ the APPNP model as the GNN backbone and utilize GAS as the baseline, given its proven superiority among all baselines. We keep all other hyperparameters consistent for a fair comparison and only vary the number of propagation layers. To demonstrate the enhanced capability of our approach in handling staleness, we specifically choose to use 5 clusters (out of a total of 40 clusters) for this ablation study, which corresponds to a high staleness situation. From Table 10, we can readily observe that the performance of both GAS and our algorithm drops after stacking more layers. This result verifies the conclusion in Theorem 3.1: the approximation error between the historical embeddings and the full batch embeddings at each layer accumulates. However, the performance of

GAS deteriorates significantly when adding more layers. In contrast, our model is not impacted to the same extent since we significantly reduce the approximation error at each layer. This demonstrates another advantage of our algorithm when deeper GNNs are utilized.

Table 10: Prediction accuracy (%) with a varying number of propagation layers on ogbn-arxiv.

| Layers | GAS | GAS+REST | GAS+REST-IS |
|--------|-----|----------|-------------|
| L=1 | 68.2 | 71.9 | 72.0 |
| L=3 | 69.4 | 71.8 | 72.3 |
| L=5 | 70.1 | 72.5 | 72.5 |
| L=8 | 69.4 | 72.4 | 72.4 |
| L=10 | 69.4 | 72.4 | 72.3 |

## J. More analysis on GraphFM

**Approximation errors** In this section, we provide more comprehensive analysis on another important baseline, GraphFM. Regarding approximation errors in Figure 18 and 19, GraphFM also experiences staleness accumulation between each layer. While it can alleviate the approximation error introduced by staleness, it incurs additional approximation error by using biased one-hop neighbors for the momentum combination. This occurs because these nodes lose aggregations from their out-of-batch neighbors, exacerbating the overall approximation error.

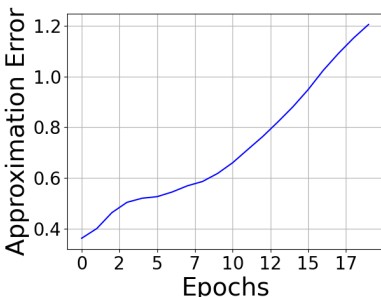

Figure 18: ogbn-arxiv

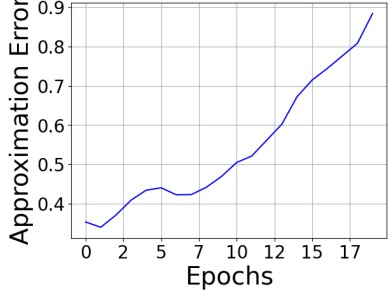

Figure 19: ogbn-products

**Performance and frequency** We provide in Table 11 for the performance of GraphFM+REST. Compared with GAS, GraphFM shows a slight performance improvement by reducing staleness, but it still falls short of achieving superior performance as it doesn't fully address the staleness issue at its source, unlike REST. Conversely, our proposed method can be seamlessly applied to GraphFM, yielding even better performance, underscoring the generality of REST. Furthermore, the frequency analysis in Table 12 indicates that higher frequencies tend to enhance performance, consistent with our claim in the main paper.

## K. Additional results comparing with SAGE

**Convergence** To better illustrate that REST outperforms not only other historical embedding methods but also classical sampling methods such as GraphSAGE, besides performance reported in Table 1, we further include Figures 20 and 21, showcasing convergence curves on ogbn-arxiv with both small and large batch sizes, utilizing time as the unit of measurement. The results indicate that our model's convergence is nearly equivalent to that of GraphSAGE. However, it's important to note that REST outperforms GraphSAGE in terms of performance and only requires a much smaller memory cost. Hence, REST demonstrates its advantage over GraphSAGE.

Table 11: Accuracy (%) improvement for GraphFM.

| DATASET | BACKBONE | PARTS | BATCH SIZE | FM | +REST | +REST-IS |
|---|---|---|---|---|---|---|
| **ogbn-products** | **GCN** | **70** | 5 | 76.3 | 77.9 | 78.0 |
| | | | 10 | 76.9 | 79.9 | 78.8 |
| | **APPNP** | **40** | 5 | 76.2 | 80.2 | 80.6 |
| | | | 10 | 77.1 | 80.3 | 80.6 |
| | **GCNII** | **150** | 5 | 75.3 | 76.2 | 76.6 |
| | | | 20 | 77.4 | 80.2 | 80.0 |
| **ogbn-arxiv** | **GCN** | **80** | 5 | 68.5 | 71.8 | 72.0 |
| | | | 10 | 70.5 | 72.0 | 72.4 |
| | | | 20 | 70.9 | 72.2 | 72.5 |
| | | | 40 | 71.8 | 72.5 | 72.7 |
| | **APPNP** | **40** | 5 | 70.3 | 72.0 | 72.4 |
| | | | 10 | 70.5 | 72.2 | 72.4 |
| | | | 20 | 71.5 | 72.3 | 72.3 |
| | **GCNII** | **40** | 5 | 70.6 | 72.7 | 72.8 |
| | | | 10 | 72.0 | 72.7 | 72.8 |
| | | | 20 | 73.1 | 73.2 | 73.1 |

Table 12: Frequency analysis of REST+GraphFM on ogbn-products.

| **Dataset** | **Freduency** | ACC |
|---|---|---|
| ogbn-products | 2 | 80.0 |
| | 3 | 80.1 |
| | 4 | 80.4 |
| | 5 | 80.6 |

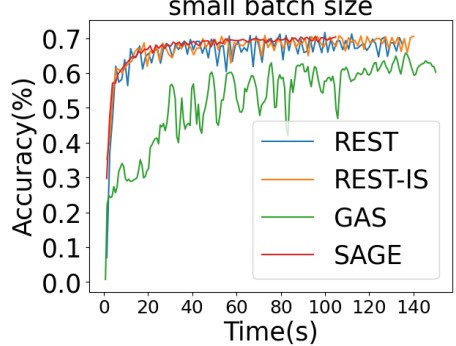

Figure 20: Small batch size

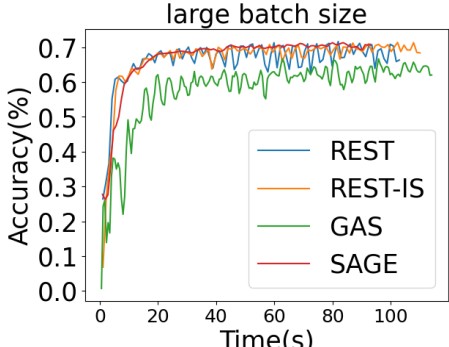

Figure 21: Large batch size

## L. Difference between REST and variation of learning rate

While there are some engineering techniques proposed to reduce feature staleness, such as adding regularization (Fey et al., 2021) or reducing the learning rate, they often suffer from many potential issues during the training process. Taking reduced learning rate as an example:

(1) Reduced convergence speed and increased training time: Lowering the learning rate can slow down convergence and prolong training time significantly.

(2) Risk of getting stuck in local minima: Lower learning rates may cause the model to get trapped in local minima for extended periods, hindering overall optimization.

(3) Sensitivity to other hyperparameters: The effectiveness of reduced learning rates can depend heavily on other hyperpa-

rameter choices, such as the optimizer used.

In contrast, REST does not encounter the optimization issues associated with engineering techniques like reducing the learning rate. It involves only additional forward passes without changing the optimization process. Furthermore, REST is highly versatile: it can utilize higher frequencies or larger batch sizes to refresh the memory bank more frequently, thereby reducing staleness. Additionally, it can incorporate importance sampling to prioritize the refreshment of embeddings for important nodes, such as REST-IS. It's important to note that REST can achieve these outcomes that traditional engineering techniques cannot accomplish.

To further support our stance, we provide a comparison between REST with different frequencies (F) and simply reducing the learning rate by F. The original learning rate used in our experiment is 0.001. We provide performance, efficiency and convergence curve in Table 13 and Figures 22, 23 to comprehensively support our claim. Upon examining the results, it's clear that when comparing F=2 with LR=0.0005 and F=5 with LR=0.0002, REST demonstrates superior performance and faster convergence rates compared to solely reducing the learning rate.

Table 13: Memory usage (MB) and running time (seconds) ogbn-products.

| **ogbn-products** | hyperparameters | ACCURACY | TIME(S) |
|---|---|---|---|
| learning rate | 0.0002 | $79.7 \pm 0.15$ | 2275 |
| Frequency | 5 | $80.5 \pm 0.09$ | 1204 |
| learning rate | 0.0005 | $79.7 \pm 0.18$ | 1950 |
| Frequency | 2 | $80.2 \pm 0.11$ | 1053 |

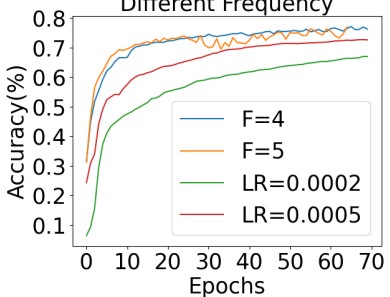

Figure 22: Convergence w.r.t epochs.

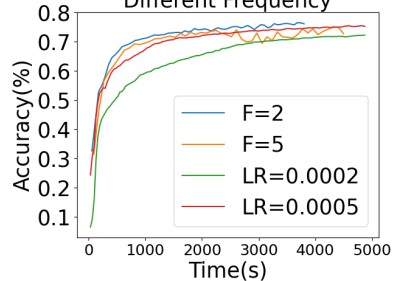

Figure 23: Convergence w.r.t time.

## M. Related Work

In this section, we summarize related works on the scalability of large-scale GNNs with a focus on sampling methods.

**Vanilla sampling methods.** Sampling methods involve dropping nodes and edges through the adoption of mini-batch training strategies, effectively reducing computation and memory requirements. In *node-wise sampling*, a fixed number of neighbors are sampled instead of considering all of them, such as GraphSAGE (Hamilton et al., 2017), PinSAGE (Ying et al., 2018) and GraphFM-IB (Yu et al., 2022). However, these methods cannot eliminate but still grapple with the neighbor explosion problem and introduce bias and variance.

*Layer-wise sampling* fixes the sampled neighbors per layer to avoid the neighbor explosion problem. For instance, FastGCN (Chen et al., 2018) independently samples nodes in each GNN layer using importance sampling. LADIES (Zou et al., 2019) onsiders the correlation between layers. ASGCN (Huang et al., 2018) further defines the sampling probability of lower layers based on the upper ones. However, the layer-wise induced adjacency matrix is usually sparser than the others, contributing to its sub-optimal performance.

Different from the previous methods, *subgraph sampling* involves sampling subgraphs from the entire graph as mini-batches and then constructing a full GNN on those subgraphs. This approach can address the neighbor explosion problem, as only the nodes within the subgraph participate in the computation. For example, ClusterGCN (Chiang et al., 2019) first

partitions the graph into clusters and then selects a subset of clusters to construct mini-batches. GraphSaint (Zeng et al., 2020) proposes importance sampling to construct mini-batches through different samplers. However, this method may introduce significant variance because the edges between subgraphs are ignored.

**Historical embedding methods.** Some advanced algorithms attempt to use historical embeddings as approximate embeddings instead of true embeddings from the full batch computation. This approach can reduce memory costs by decreasing the number of sampled neighbors, either per hop (Chen et al., 2017) or in terms of the number of hops (Fey et al., 2021). VR-GCN (Chen et al., 2017) first proposed this idea of restricting the number of sampled neighborhoods per hop and using historical embeddings for out-of-batch nodes to reduce variance. MVS-GCN (Cong et al., 2020) simplified this scheme into a one-shot sampling scenario, where nodes no longer need to recursively explore neighborhoods in each layer. GNNAutoScale (Fey et al., 2021) further restricts the neighbors to their direct one-hop but without discarding any data, enabling it to maintain constant GPU memory consumption. GraphFM-OB (Yu et al., 2022) employs feature momentum to further improve performance.

Although these historical embedding approaches are promising because of their scalability and efficiency, they all suffer from approximation errors originating from feature staleness. This issue has become a bottleneck, especially for large-scale datasets, as demonstrated in our preliminary study in the main paper. Several works have proposed techniques to reduce staleness. For instance, GAS (Fey et al., 2021) concludes the staleness is from the inter-connectivity between batches and uses graph clustering to relieve it and also uses regularization to prevent model parameters change too much. GraphFM-OB (Yu et al., 2022) uses feature momentum of in-batch and out-of-batch nodes for compensation. Refresh (Huang et al., 2023) alleviates the staleness issue by establishing staleness criteria. However, these approaches fail to address the fundamental issue, which is the discrepancy in updating frequency between memory tables and model parameters. Besides, their methods are complementary to ours and can be seamlessly incorporated into our algorithm.

## N. Differences between REST and Gradient Accumulation

REST aims to refresh the memory bank to reduce staleness, whereas gradient accumulation simulates the effect of increasing the batch size. Hence, gradient accumulation introduces additional computation overhead. In terms of runtime, it prolongs the time since every forward pass requires gradient computation. Regarding memory usage, gradients must be stored for each forward pass until backward propagation is executed with accumulation. This accumulation of gradients across multiple forward passes can result in increased memory consumption, particularly with large models or datasets. To demonstrate the efficiency contrast between REST and gradient accumulation, we present the following Table 14, showcasing the difference and superior efficiency of our design. Note that REST can also be combined with gradient accumulation since they are orthogonal techniques. We leave this as a future work.

Table 14: Memory usage (MB) and running time (seconds) on ogbn-products.

|  | MEMORY(MB) | TIME/EPOCH(S) |
| --- | --- | --- |
| REST | 9295 | 33 |
| Gradient Accumulation | 14537 | 39 |

## O. Minor Baselines

In addition to major baselines such as GAS, GraphFM, LMC, and Refresh, there are also several minor baselines that address the issue of optimizing staleness. For example, S3 (Wang et al., 2024) uses staleness scores to guide the sampling process, and SAT (Bai et al., 2023) alleviates staleness through distributed training. We use GCN as GNN backbone. The results are shown in Table 15:

From the results, we can see that REST easily outperforms these baselines without requiring a complicated design. It is important to note that their approaches are entirely orthogonal and can be combined with ours, given that our method is quite general. However, since their code is not yet available, we only report the numbers from their respective papers for comparison purposes.

Table 15: Performance Comparison with Minor Baselines.

| Models | OGBN-ARXIV | OGBN-PRODUCTS |
|--------|------------|---------------|
| S3 | 72.1 | 77.2 |
| SAT | 72.0 | 78.9 |
| REST | **73.2** | **80.5** |

## P. Broad Impact and Generalizability

Besides alleviating the staleness issue, REST also addresses the broader issue of staleness caused by asynchronous updates—a common challenge in many settings. To highlight our contributions, we present several additional scenarios in which REST can be applied:

- Distributed Training: Asynchronous updates often arise in distributed training due to communication overhead. Nodes may operate on potentially stale global parameters from central servers, leading to parameter staleness. Applying REST in this scenario means performing extra forward passes without gradient calculation asynchronously, which refreshes local embeddings (or activation caches) more frequently. Consequently, when the node eventually computes gradients or synchronizes with the central server, the resulting update is less affected by staleness, improving overall convergence.

- Federated Learning: Federated learning suffers from parameter staleness due to infrequent client–server communications. REST's decoupling strategy can be applied by letting the clients (or server) perform additional forward-only steps between global synchronizations. These extra forward passes keep local representations up to date with the client's current model version, so that when the client finally computes gradients and communicates them back, the cached embeddings are no longer heavily stale. These additional asynchronous forward updates serve as opportunities for refreshing beyond global synchronization. This mitigates the mismatch arising from stale parameters and promotes faster convergence in federated learning.

This broader applicability indicates REST is not merely a method specific to historical embeddings, but a generally beneficial framework for addressing asynchronous update issues widely prevalent in machine learning training.

## Q. Difference from Related Works

As emphasized in the main text, GAS and related methods are specialized, scalable historical-embedding techniques: they apply a fixed training optimization to improve the scalability of GNN backbones (e.g., GCN) and reduce bias from traditional sampling methods. In contrast, REST is a novel, general training framework that effectively alleviates the most notable staleness issue at its root in all such approaches, offering a fundamentally new training optimization strategy to eliminate the staleness introduced by GAS and similar approaches—thereby significantly boosting performance and accelerating convergence. In GAS, the memory table updates only the embeddings of nodes that appear in the current batch. As a result, most node embeddings remain stale because GAS and similar approaches inevitably suffer from a frequency mismatch between cache updates and model parameter updates, leading to persistent staleness. Our work addresses this root cause by proposing a general training framework that ensures more frequent—and more flexible—refreshing of historical embeddings to eliminate staleness without changing the model architecture, aggregator, or sampling algorithm, rather than merely designing a model that differs from GAS. Meanwhile, it can be seamlessly integrated with any historical embedding method.

## R. Hyperparameters Searching Space

Our model's hyperparameters are tuned from the following search space: (1) learning rate: $\{0.01, 0.001, 0.005\}$; (2) weight decay: $\{0, 5e-4, 5e-5\}$; (3) dropout: $\{0.1, 0.3, 0.5, 0.7, 0.8\}$; (4) propagation layers : $L \in \{2, 3, 4\}$; (5) MLP layers: $\{3, 4\}$; (6) MLP hidden units: $\{128, 256, 512\}$.

