# OpenReview forum: "Haste Makes Waste: A Simple Approach for Scaling Graph Neural Networks"
_ICML.cc/2025/Conference — ICML 2025 poster_

### Official Review · Reviewer_y9S9 · 2025-03-09

**Overall Recommendation:** 3

**Summary:**

This paper proposes a simple yet highly effective training algorithm (REST) to effectively reduce feature staleness. The proposed REST significantly improves performance and convergence across varying batch sizes, especially when staleness is predominant. Experiments demonstrate that REST achieves a 2.7% and 3.6% performance enhancement on the ogbn-papers100M and ogbn-products dataset.


## update after rebuttal


I thank the authors for the authors' rebuttal. I would like to keep my original evaluations.

**Claims And Evidence:**

Theorem 3.1 gives the approximation errors by Equation (5). However, Equation (5) may not hold, as the assumption of Lipschitz continuity about  $\nabla_{\theta} L$ is not reasonable. Consider a two-layer GCN message passing on a graph $(\\{v\_1,v\_2\\}, \\{ (v_1,v_2) \\})$. The message passing is $h\_1^{(2)}=W\_2h\_2^{(1)}=W\_2\sigma(W\_1h\_1^{(0)})$. We construct two subgraphs $S\_1,S\_2$ induced by $\{v\_1\}$ and $\{v\_2\}$ respectively. Suppose that the historical embeddings are equal to true embeddings, i.e., ${\hat{h}}\_i^{(j)}=h\_i^{(j)}$. The gradients with the historical embeddings are $\nabla_{W\_1}L({\hat{h}}\_1^{(2)})=\nabla_{W\_1}{\hat{h}}\_2^{(1)} \nabla_{{\hat{h}}\_2^{(1)}}{\hat{h}}\_1^{(2)} \nabla_{\hat{h}\_1^{(2)}}L({\hat{h}}\_1^{(2)})=\nabla_{W\_1}{\hat{h}}\_2^{(1)} \cdot 0 \cdot \nabla_{\hat{h}\_1^{(2)}}L({\hat{h}}\_1^{(2)})=0$, while the gradients of the  true embeddings $\nabla_{{h}\_2^{(1)}}{h}\_1^{(2)} \neq 0 $.

**Essential References Not Discussed:**

The cited references are sufficient.

**Experimental Designs Or Analyses:**

The authors may want to report the standard deviation in Table 1.

**Methods And Evaluation Criteria:**

The authors propose REST to reduce feature staleness, which is useful to accelerate convergence.

**Other Comments Or Suggestions:**

See Weaknesses.

**Other Strengths And Weaknesses:**

Strengths:

1. The proposed REST is simple yet highly effective.
2. Experiments demonstrate strong scalability of REST to large-scale datasets, such as ogbn-papers100M, ogbn-products, and MAG240M.



Weaknesses:

1. Equation (5) may not hold, as the assumption of Lipschitz continuity about  $\nabla_{\theta} L$ is not reasonable. Consider a two-layer GCN message passing on a graph $(\\{v\_1,v\_2\\}, \\{ (v_1,v_2) \\})$. The message passing is $h\_1^{(2)}=W\_2h\_2^{(1)}=W\_2\sigma(W\_1h\_1^{(0)})$. We construct two subgraphs $S\_1,S\_2$ induced by $\{v\_1\}$ and $\{v\_2\}$ respectively. Suppose that the historical embeddings are equal to true embeddings, i.e., ${\hat{h}}\_i^{(j)}=h\_i^{(j)}$. The gradients with the historical embeddings are $\nabla_{W\_1}L({\hat{h}}\_1^{(2)})=\nabla_{W\_1}{\hat{h}}\_2^{(1)} \nabla_{{\hat{h}}\_2^{(1)}}{\hat{h}}\_1^{(2)} \nabla_{\hat{h}\_1^{(2)}}L({\hat{h}}\_1^{(2)})=\nabla_{W\_1}{\hat{h}}\_2^{(1)} \cdot 0 \cdot \nabla_{\hat{h}\_1^{(2)}}L({\hat{h}}\_1^{(2)})=0$, while the gradients of the  true embeddings $\nabla_{{h}\_2^{(1)}}{h}\_1^{(2)} \neq 0 $.
2. The authors may want to report the standard deviation in Table 1.

**Questions For Authors:**

See Weaknesses.

**Relation To Broader Scientific Literature:**

The related work mainly focuses on graphs with less than three million nodes (e.g., ogbn-products) [Ref1][Ref2], while the proposed REST can scale to ogbn-papers100M and MAG240M, which contain at least one hundred million nodes.



[Ref1] Gnnautoscale: Scalable and expressive graph neural networks via historical embeddings. ICML 2021.


[Ref2] Lmc: Fast training of gnns via subgraph sampling with provable convergence. ICLR 2023.

**Theoretical Claims:**

1. Equation (5) may not hold, as the assumption of Lipschitz continuity about  $\nabla_{\theta} L$ is not reasonable. Consider a two-layer GCN message passing on a graph $(\\{v\_1,v\_2\\}, \\{ (v_1,v_2) \\})$. The message passing is $h\_1^{(2)}=W\_2h\_2^{(1)}=W\_2\sigma(W\_1h\_1^{(0)})$. We construct two subgraphs $S\_1,S\_2$ induced by $\{v\_1\}$ and $\{v\_2\}$ respectively. Suppose that the historical embeddings are equal to true embeddings, i.e., ${\hat{h}}\_i^{(j)}=h\_i^{(j)}$. The gradients with the historical embeddings are $\nabla_{W\_1}L({\hat{h}}\_1^{(2)})=\nabla_{W\_1}{\hat{h}}\_2^{(1)} \nabla_{{\hat{h}}\_2^{(1)}}{\hat{h}}\_1^{(2)} \nabla_{\hat{h}\_1^{(2)}}L({\hat{h}}\_1^{(2)})=\nabla_{W\_1}{\hat{h}}\_2^{(1)} \cdot 0 \cdot \nabla_{\hat{h}\_1^{(2)}}L({\hat{h}}\_1^{(2)})=0$, while the gradients of the  true embeddings $\nabla_{{h}\_2^{(1)}}{h}\_1^{(2)} \neq 0 $.

---

> ### Author Rebuttal · Authors · 2025-04-01
>
> **Anonymous link: https://anonymous.4open.science/r/REST_ICML2025-0972/REST_ICML_2025_Reviewer_y9S9.pdf**
>
> **A1** Counterexample of Theorem 3.1
>
> We sincerely appreciate the reviewer's careful and thoughtful comments, and we are happy to clarify this counterexample.
>
> Based on our understanding of the review, a special graph was constructed with only two nodes ($v_1, v_2$), where each induced subgraph contains exactly one node. Under this extreme scenario, each subgraph includes only the node itself. The reviewer further assumes that the historical embeddings exactly match the true embeddings, which leads to the gradient becoming zero under these special conditions:
>
> $\nabla_{\hat{h}_2^{(1)}}\hat{h}_1^{(2)}= 0$
>
> In the full batch training scenario, due to the existence of actual message passing, we have:
>
> $\nabla_{h_2^{(1)}} h_1^{(2)} \neq 0$
>
> Thus, the statement argues that the assumption of Lipschitz continuity may not hold.
>
> However, we would like to clearly point out that this scenario does **not** align with the historical embedding or neighbor aggregation logic used in our method (and in the broader family of historical embedding methods):
>
> Firstly, we want to clarify the definition of "out-of-batch" nodes commonly used in all historical embedding methods. Existing methods typically first sample a mini-batch (or subgraph) of nodes termed "in-batch" nodes, whose embeddings are updated every iteration. Then, they select all direct one-hop neighbors of these "in-batch" nodes as "direct one-hop out-of-batch" nodes (simply referred to as "out-of-batch" nodes in existing literature) and use their historical embeddings to approximate the true embeddings during aggregations to reduce computation bias. All other nodes that are not directly connected to the batch are simply discarded and referred to as "other out-of-batch nodes." Therefore, since the message passing is given by
> $h_1^{(2)} = W_2 h_2^{(1)} = W_2 \sigma(W_1 h_1^{(0)})$, this indicates that there is an edge between $v_1$ and $v_2$.
> if we consider $v_1$ as an in-batch node, then $v_2$ should exactly be treated as a "direct one-hop out-of-batch node" whose historical embedding is used in aggregation—even though they are not in the same subgraph—rather than simply discarding it as an "other out-of-batch node." We also provide a figure in the anonymous link that better explains the definition.
>
>
> Specifically, in most historical embedding methods, the absence of node $v_2$ explicitly from the subgraph does **not** imply the complete loss of its influence on node $v_1$. Instead, historical embedding methods utilize historical embeddings $\hat{h}_2^{(1)}$ to preserve neighbor information, and gradients in the backward pass will still propagate through these stored embeddings. For "out-of-batch" nodes, their previously stored activation values are retrieved from memory to ensure their influence can still be incorporated. A typical form can be represented as:
>
> $h_i^{(l)} = \text{Agg}\left(\\\{h_j^{(l-1)}\mid j \in \text{in-batch}\\\} \cup \\\{\hat{h}_k^{(l-1)}\mid k \in \text{out-of-batch}\\\}\right)$
>
> Thus, the update of $v_1$'s embeddings still references $\hat{h}_2^{(1)}$, even if $v_2$ is not explicitly included in the sampled mini-batch/subgraph. Consequently, node $v_1$ will indeed incorporate $\hat{h}_2^{(1)}$, preserving
>
> $\nabla_{\hat{h}_2^{(1)}}(\hat{h}_1^{(2)})\neq 0$,
>
> and $\nabla_{W_1}L$ does **not** simply vanish but instead includes the gradient contributions from $\hat{h}_2^{(1)}$. Thus, the proposed counterexample does not hold under historical embedding methods.
>
> Moreover, **the Lipschitz continuity assumption regarding the gradients is a general and widely accepted assumption in the literature, such as in LMC[1].** It has consistently proven valid empirically across a wide range of realistic datasets and practical training scenarios.
>
> **A2** The standard deviation in Table 1.
>
> We conducted multiple runs and included the standard deviation in Table 1. Please see the updated table in the anonymous link.
>
> We appreciate your feedback and have worked diligently to address all your concerns. If you have any further questions, please let us know, and we kindly request that you consider adjusting the scores in light of our revisions.
>
> [1] LMC: Fast Training of GNNs via Subgraph Sampling with Provable Convergence

---

> > ### Comment · Reviewer_y9S9 · 2025-04-02
> >
> > Thanks for your rebuttal. My remaining concerns are as follows.
> >
> > 1. Does REST compute $\nabla\_{W\_1} \hat{h}\_2^{(1)}$? In my opinion, as the historical embedding $\hat{h}\_2^{(1)}$ is directly pulled from an offline storage, the forward pass $\hat{h}\_2^{(1)}=\sigma(W\_1 \hat{h}\_1^{(0)})$ and the corresponding backward pass are missing. If $\nabla\_{W\_1} \hat{h}\_2^{(1)}=0$, then $\nabla\_{W\_1} L(\hat{h}\_1^{(2)}) \neq \nabla\_{W\_1} L(h\_1^{(2)})$.
> > 2. In LMC [1] and VR-GCN [2], the Lipschitz continuity assumption is made with respect to the gradients of the GNN parameters (i.e. $W\_1$ and $W\_2$ in the example) rather than the GNN embeddings (i.e. $h\_i^{(1)}$ and $h\_i^{(2)}$ in the example).
> >
> >
> > [2] Stochastic Training of Graph Convolutional Networks with Variance Reduction.

---

> > > ### Author Response · Authors · 2025-04-03
> > >
> > > We appreciate your quick reply, and we are happy to answer all your further questions.
> > >
> > > **A1**
> > > First, we would like to note that the proposed REST is a general training framework designed to address the staleness issue inherent in all related works by decoupling forward and backward propagation, so **the gradient flow during the backprop entirely follows that of the backbone historical embedding models** (e.g., GAS, GraphFM, and LMC). This design renders our model general and applicable to any existing approach. Therefore, whether $\nabla_{W_1} \hat{h}_2^{(1)}$ is computed depends entirely on the backbone historical embedding model with which REST is combined, rather than on REST itself.
> > >
> > > Nonetheless, we are happy to provide additional details on this matter.
> > > Consider the gradient of the loss $L$ with respect to $W_1$ expanded via the chain rule:
> > > $$
> > > \nabla_{W_1} L = \sum_{i\in B}\sum_{j \in N(i)} \frac{\partial L}{\partial h_i^{(2)}} \frac{\partial h_i^{(2)}}{\partial h_j^{(1)}} \frac{\partial h_j^{(1)}}{\partial W_1}
> > > $$
> > >
> > > For GAS and GraphFM, only the in-batch nodes participate in gradient computation and backprop. Since the historical embeddings from each layer are retrieved from a cache, the backward prop cannot flow through these cached values. Hence, the gradients with respect to the historical embeddings satisfy $\nabla_{W_1} \hat{h}_2^{(1)} = 0 $ and they do not affect the model parameter updates.
> > >
> > > In other words, $\nabla_{W_1} L(\hat{h}_1^{(2)})$  is not equal to
> > >
> > > $\nabla_{W_1} L(h_1^{(2)})$ ,
> > >
> > > which is the gradient bias introduced by GAS and GraphFM.
> > >
> > > Formally,  in GAS and GraphFM, for the term $\frac{\partial h_j^{(1)}}{\partial W_1}$, if node $j$ is an in-batch node, then $h_j^{(1)} $ requires a gradient and participates in backprop, so $\frac{\partial h_j^{(1)}}{\partial W_1} \neq 0$,
> > > However, if $j$ is an out-of-batch neighbor (its embedding is cached as $ \hat{h}_j^{(1)} $), then $\frac{\partial \hat{h}_j^{(1)}}{\partial W_1} = 0$.
> > >
> > > **In contrast**, for LMC, this drawback of losing the gradient for historical embeddings in GAS motivates their approach. It maintains a memory table for the historical gradients and explicitly retrieves and compensates for the discarded messages during backprop. By employing a gradient momentum technique, LMC proactively compensates for the gradients of out-of-batch nodes, thereby avoiding the loss of these gradients (Equation 12 in the paper). Consequently, $ \nabla_{W_1} \hat{h}_2^{(1)} $ is maintained and approximated. In other word, during the backprop, LMC maintains the gradient dependencies related to $ \hat{h}_2^{(1)} $ using the proposed compensation formulas (Equations 11–13 and Figure 1), which significantly reduces the gradient bias and provides convergence guarantees (Theorems 2 and 3). This is why LMC achieves accurate gradients and better performance. Therefore, if apply REST to LMC, that gradient is still preserved (Appendix E)
> > >
> > > In REST, we focus on the root cause of staleness as detailed in Section 2. The improved performance and accelerated convergence observed with GAS in the main text, GraphFM and LMC in Appendix I and E demonstrate that REST is applicable to any scenario—such as the two cases regarding gradient flow discussed above—without requiring modifications to the workflow of the underlying model, thereby highlighting the generalizability and novelty of our approach.
> > >
> > > **A2**  We first note that two papers adopt different analytical perspectives
> > > (1) In LMC, to analyze convergence and how the gradients change with respect to the parameters $\theta$, a Lipschitz assumption evaluated on the gradients with parameters is made.
> > > (2) In REST, to analyze how the discrepancy between the historical and true embeddings affects the final gradient, we make a Lipschitz assumption on the gradient evaluated on embeddings $h$.
> > >
> > > Secondly, we emphasize that our assumption is also reasonable since several key components of GNNs satisfy Lipschitz continuity. Consider a GNN with $L$ layers, a typical layer can be expressed as
> > > $$
> > > h^{(\ell+1)}_v = \text{UPDATE}\Bigl( h^{(\ell)}_v, \; \text{AGG}\bigl(\{h^{(\ell)}_u : u \in \mathcal{N}(v)\}\bigr)\Bigr).
> > > $$
> > >
> > > Then, the following assumptions usually hold in the literature (e.g., in GAS):
> > >
> > > (1) The aggregation function evaluated on the embeddings, $\text{AGG}$, is $\alpha$-Lipschitz, where functions such as mean, sum, and max are all Lipschitz continuous.
> > >
> > > (2) The update function evaluated on the embeddings, $\text{UPDATE}$, is $\beta$-Lipschitz.
> > >
> > > Furthermore, common activation functions and loss functions with respect to the embeddings are also Lipschitz continuous. When these components are composed to form the full GNN, the overall Lipschitz continuity is preserved. Consequently, the gradient evaluated on the embeddings is Lipschitz continuous. This assumption is standard and reasonable in the analysis of GNNs, particularly when investigating how errors in the node embeddings affect the overall gradient estimation.

---

### Official Review · Reviewer_1EQa · 2025-03-12

**Overall Recommendation:** 2

**Summary:**

This paper presents a simple yet effective training approach called REST for scaling GNNs. The authors analyze the issue of embedding staleness in historical embedding methods, demonstrating that stale features negatively impact model convergence and performance. REST addresses this issue by decoupling forward and backward propagations and adjusting their execution frequencies, significantly reducing feature staleness. Experimental results indicate that REST achieves superior performance and faster convergence on several large-scale benchmark datasets.

##Update after rebuttal

In the 2nd-round rebuttal, the authors provided more clearer differences between their work and existing works, and hence I changed my mind from "weak reject" to "neutral". I will respect AC or other reviewers to further discuss the outcome.

**Claims And Evidence:**

The proof of Theorem 3.2 is missing. Hence the claim “REST achieves a faster convergence rate theoretically” is not convincing.

**Essential References Not Discussed:**

The related works are discussed in the appendix. However, the most related works such as GAS are not discussed in detail in the main context. Consequently, the differences between the proposed work and previous works are not well explained and compared.

**Experimental Designs Or Analyses:**

Yes. The experiments are designed scientifically and the results can support the claim.

**Methods And Evaluation Criteria:**

Yes

**Other Comments Or Suggestions:**

None

**Other Strengths And Weaknesses:**

Strengths:
1.	The experiments are well set, and the results seem promising with respect to accuracy.
2.	The issues in the scalable GNN are well-targeted. In other words, the motivation of this paper is built properly.
Weakness:
1.	This paper should add more explanation to some specific phrases. For example, this paper can provide the definition for the words such as “in-batch” and “out-batch”. The lack of explanation will make this paper hard to follow.
2.	The novelty of this paper is not properly highlighted. It seems that this paper incorporates many insights from the paper of GAS (Fey et al., 2021), including the reference of Theorem 3.1. From my point of view, this paper only modifies a minor implementation detail of GAS and has not proposed enough theoretical contribution. In order to highlight the main contribution, this paper should provide a detailed introduction of strongly related works and compare them theoretically.
3.	The detailed proof of theorem 3.2 is missing.

**Questions For Authors:**

Please see the weakness part.

**Relation To Broader Scientific Literature:**

Starting from the scalable issues in GNN, this paper builds its contribution on the drawbacks of other GNN methods which utilize the historical embeddings. It also highlights its novelty in directly addressing the embedding staleness issue at its root by decoupling forward and backward propagations.

**Theoretical Claims:**

The proof of Theorem 3.2 is missing. Although there are some references proving the upper bound of the expectation of gradients’ norm, its complex form seems not trivial to prove. A more detailed or self-contained proof should be provided.

---

> ### Author Rebuttal · Authors · 2025-04-01
>
> **Anonymous link**: https://anonymous.4open.science/r/REST_ICML2025-0972/REST_ICML_2025_Reviewer_1EQa.pdf
>
> **A1** Proof of Theorem 3.2
>
> Please check the detailed proof in the anonymous link . We would like to clarify that the main result in Theorem 3.2 shares foundational assumptions with previous works, which we explicitly cited in line 253. To avoid misinterpretation, we initially chose to cite the references rather than showing our proofs in submission.
>
> **A2** Difference between GAS and REST
>
> We believe that the essential technical aspects of GAS are adequately covered in **Section 2.1**. Given the space constraints of the rebuttal, please refer to our main text, and we are happy to address any further questions you may have.
>
> Difference: **Overall, GAS and other related works are specific historical embedding methods, whereas REST is a novel, general training framework that effectively solves the most notable staleness issue in these methods, boosting performance and accelerating convergence.** In GAS, most node embeddings remain stale because these methods inherently suffer from a frequency mismatch between cache updates and model parameter updates. Our work tackles this issue by introducing a general training framework that can be integrated with any related approach, enabling more frequent refreshing of historical embeddings and thereby eliminating staleness without altering the model architecture—rather than merely designing a model that differs from GAS.
>
> **Thus, the target of our work is not the introduction of a new model, but the development of a widely applicable training framework that addresses a key limitation—staleness—in all related works. Moreover, it provides promising solutions for other tasks involving asynchronous update scenarios in machine learning.**
>
> **A3** Specific Phrases
>
> In this work, we follow the definitions used in existing related works. Due to space constraints, we did not reiterate these definitions in our submission. However, please refer to the figure in the anonymous link for a more visual explanation.
>
> **A4** Contribution Highlight
>
> We would like to respectfully disagree with the statement. We hope to clarify any misunderstandings regarding our objectives and highlight the significant novelty and contributions:
>
> First, the main focus of our work is **not** to introduce a new historical embedding model architecture, but rather to **propose a general, effective, efficient, and simple training framework that fundamentally resolves the staleness issue present in all existing  methods.** In other word, our model aims to address the critical shortcomings in GAS and other related works, rather than designing a model that differs from GAS. **The requirement of minor implementation modification precisely demonstrates our model's strength rather than its weakness.** REST exhibits strong generalizability and can significantly improve performance and accelerate convergence by eliminating staleness in historical embeddings, without requiring any changes to the underlying model.
>
> Second, our work is the first to provide a comprehensive analysis of staleness and to reveal that a mismatch in update frequencies is its root cause—a research aspect that has not been explored before. We not only employ Theorem 1 to directly link the frequency mismatch to staleness, but also present extensive empirical results demonstrating that existing methods inevitably suffer from severe staleness issues.
>
> Third, REST introduces a previously unexplored method: decoupling forward and backward prop so that the memory table can refresh at a frequency different than the parameter updates. **To the best of our knowledge, this core innovation does not appear in GAS or any other existing method. Rather than merely being a modification, our approach can be viewed as a more general form of GAS.** Our experiments demonstrate substantial performance improvements as well as significantly faster convergence. These empirical results not only validate our theoretical claims but also highlight practical advantages that GAS does not achieve. Moreover, this idea can be applied to any staleness issue arising from asynchronous updates—a common challenge in various machine learning tasks—providing promising solutions for future work that hold significant value for the entire community. Furthermore, we introduce a novel variant, REST-IS, which employs a unique importance sampling strategy to further mitigate staleness, showcasing additional methodological innovation.
>
> Fourth, Theorem1 in our paper diverges from GAS. We also introduce Theorem 3.2, which substantiates the empirical observation that REST achieves faster convergence. This rigorous theoretical contribution clearly differentiates our work from GAS. Please refer to the anonymous link for proof.
>
> We believe our work offers a more substantial advancement and provides promising solutions for future research, rather than merely proposing a specific model design.

---

> > ### Comment · Reviewer_1EQa · 2025-04-03
> >
> > Thanks for your explanation and I understand better of your work's position.
> >
> > In your rebuttal, you emphasize that “GAS and other related works are specific historical embedding methods, whereas REST is a ... framework". However, GAS is also self-described as a framework and its experiments integrate various existing models, blurring the distinction between the GAS and REST. Moreover, both approaches address training optimization, which may further diminish the uniqueness of your contribution. As such, I am not fully unconvinced that REST represents a significant advancement over the baseline GAS.

---

> > > ### Author Response · Authors · 2025-04-05
> > >
> > > We appreciate your reply and we want to clarity the difference between GAS and REST from both "training optimization" and "framework" mentioned in your reply.
> > >
> > > (1) Conceptually different training optimization:
> > >
> > > Characterizing both approaches simply as "training optimization" oversimplifies their distinct contributions. Under such broad categorization, most training innovations could be similarly dismissed. **REST operates at a different level than GAS - it's not an alternative to historical embedding methods but rather a complementary framework for improving them.** Specifically:
> > >
> > > **GAS introduces a fixed training optimization method aimed primarily at enhancing the scalability of GNN backbones (e.g., GCN) while mitigating bias from traditional sampling methods.** It achieves this by caching historical embeddings to reduce the variance in mini-batch approximations. However, during training, its inflexible update strategy causes significant embedding staleness—historical embeddings update far more slowly than the model parameters, creating a fundamental performance bottleneck. In other words, GAS mainly focuses on reducing memory costs and achieving better performance via historical embeddings, with staleness emerging as an unintended side effect of this and all subsequent historical embedding methods.
> > >
> > > In stark contrast, **REST provides a fundamentally novel training optimization strategy whose primary optimization goal is to exactly solve the staleness issue introduced by GAS and other historical embedding methods, explicitly addressing the root cause of embedding staleness.** We emphasize that instead of specifically focusing on modifying the computation graph during message passing by using cached embeddings like in GAS, REST focuses on general machine learning training optimization. It dynamically decouples the frequencies of forward and backward propagation, enabling frequent and independent updates. This strategy drastically reduces staleness, as clearly proven by our theoretical analysis and extensively validated by empirical results. **Moreover, REST is not limited to a fixed scalable training method like GAS; rather, it serves as a flexible and general optimization framework compatible with virtually all existing historical embedding-based methods (including GAS itself)**, significantly enhancing their performance and convergence speed. It can also be applied to scenarios with asynchronous updates, such as distributed training and federated learning, as mentioned in the rebuttal to Reviewer uPqW.
> > >
> > > In summary, **the training optimization objectives, methods, and motivations of the works are completely different. REST should be considered a complementary framework for enhancing historical embedding methods rather than a modification to them.**
> > >
> > > (2) Completely different framework scopes:
> > >
> > > We want to emphasize that the training framework also operates at a different and more general level—similar to the training optimization discussed above:
> > >
> > > In GAS, the authors describe it as a framework, meaning that it only can be combined with different GNN backbones (e.g., GCN, GCNII), which indicates that the proposed caching strategy does not require a specific message passing pattern, as observed in their experiments. However, it still relies on a specific, flawed update strategy and graph partitioning to achieve its objective, which is why we refer to it as a specific historical embedding model.
> > >
> > > In contrast, **REST is a meta-framework that operates at a higher level to improve any historical embedding method, including GAS itself. This hierarchical relationship clearly distinguishes REST from GAS.** As shown in Table 2 and 3, our model can be integrated with various historical embedding methods, different GNN backbones, and diverse sampling strategies—an approach that is entirely different from GAS and cannot be achieved by GAS.
> > >
> > > Thus, the fundamental distinction is clear: **GAS is a specific historical embedding-based scalable training method, limited by its fixed update and caching strategy, while REST is a universally applicable optimization framework providing a general and principled approach to embedding updates on any historical embedding methods, fundamentally resolving a key bottleneck (staleness) inherent in methods like GAS.** This substantial conceptual difference in flexibility, generality, and staleness-awareness decisively positions REST as a significant advancement over GAS.
> > >
> > > Based on the two points above, **GAS and REST address completely different problems, with distinct training optimization objectives and techniques. REST also operates at a higher hierarchical level than GAS with respect to training.** We believe that REST offers a broader contribution to existing work and provides a promising solution for future research.
> > >
> > > We hope we have addressed all your concerns. We kindly request that you consider adjusting your scores in light of our revisions, as this is very important to us.

---

### Official Review · Reviewer_mAEj · 2025-03-14

**Overall Recommendation:** 2

**Summary:**

The paper studies the problem of scaling the use of graph neural
networks to large graphs. Existing techniques make use of the
historical features, which may become outdated. On the contrary,
the paper introduces REST algorithm, which contains the
influence of the outdated features. Also, the new model
can merge with current pipelines and advance their
performance and the machine learning algorithm's convergence.

## Update after rebuttal: I thank the authors for their response about the F1 metric. After the extensive answers to the other reviews of the paper, I am acknowledging the deeper expertise of the other reviewers and basing my opinion on their comments.

**Claims And Evidence:**

The paper is clear to read and carefully written for ideas and their materialization to `REST Technique'.

**Essential References Not Discussed:**

After a first study, there are not any references missing.

**Experimental Designs Or Analyses:**

The authors employ extensive experiments of their
solution (`REST') with 5 datasets
and list the performing accuracy metric in them; it is
higher than that of the existing techniques (VR-GCN,
MVS-GCN, GCN, APPNP, GCNII).

**Methods And Evaluation Criteria:**

In the comparisons' two tables, the authors report the accuracy of the Graph Classification task. How about the other metrics, e.g. AUC ROC, Mean squared error, Pr/Recall, Pr/Recall AUC, F1 with 0.5 as threshold (or with varying thresholds)?

**Other Comments Or Suggestions:**

NA.

**Other Strengths And Weaknesses:**

No more comments.

**Questions For Authors:**

No more questions.

**Relation To Broader Scientific Literature:**

The suggested approach seems novel from a first overview of the GNN literature.

**Theoretical Claims:**

The paper suggests the solution of containing the
outdated features
by decoupling the forward and backward propagation
of the neural network weight updates and
dynamically adjusts their execution frequencies,
permitting the
memory table to be renewed faster than than the model
parameters.

---

> ### Author Rebuttal · Authors · 2025-04-01
>
> We sincerely appreciate your insightful comments and thoughtful questions.
>
> **A1** Other Metrics of performance:
>
> We choose to use accuracy as the metric in our submission since all other baselines use it, ensuring a fair comparison. Since the major task is a multi-class classification problem, we follow your suggestion and further include micro and macro F1 scores as additional metrics to better highlight the advantages of REST/REST-IS. Due to the limited time available for the rebuttal, we currently report the performance of state-of-the-art baselines, GAS and GraphFM on ogbn-arxiv and ogbn-products. We use GCN as our GNN backbone for all methods including REST. We will include similar results for other baselines on other large-scale datasets in our revision of the main text.
>
> From the results in the following table, we observe that REST outperforms state-of-the-art baselines across various evaluation metrics on all datasets, demonstrating superior performance in large-scale training. This is consistent with the conclusions in our main submission. Moreover, while historical embedding methods still suffer from the staleness problem—resulting in significant performance drops on large-scale datasets such as ogbn-products—REST addresses the problem at its source, delivering significant performance enhancements that further validate our main text's findings.
>
> | Models   | ogbn-arxiv Micro-F1 | ogbn-arxiv Macro-F1 | ogbn-products Micro-F1 | ogbn-products Macro-F1 |
> |----------|---------------------|---------------------|------------------------|------------------------|
> | Sage     | 71.5 ± 0.2          | 52.1 ± 0.1          | 78.7 ± 0.1             | 37.0 ± 0.1             |
> | GAS      | 71.7 ± 0.2          | 52.5 ± 0.1          | 76.7 ± 0.2             | 35.3 ± 0.1             |
> | GraphFM  | 71.8 ± 0.2          | 52.6 ± 0.2          | 76.8 ± 0.2             | 35.5 ± 0.1             |
> | REST     | 72.2 ± 0.2          | 53.3 ± 0.1          | **79.6 ± 0.1**             | **39.1 ± 0.1**             |
> | REST-IS     | **72.3 ± 0.1**          | **53.4 ± 0.1**         | 78.6 ± 0.1             | 38.6 ± 0.1             |
>
> We appreciate your feedback and have worked diligently to address all your concerns. If you have any further questions, please let us know, and we kindly request that you consider adjusting the scores in light of our revisions.

---

### Official Review · Reviewer_uPqW · 2025-03-15

**Overall Recommendation:** 3

**Summary:**

In this paper, the author proposes an algorithm to mitigate the issue of feature staleness.


## update after rebuttal
I would thank the author for the rebuttal, and my score remains the same.

**Claims And Evidence:**

The claims appear to be valid.

**Essential References Not Discussed:**

N/A

**Experimental Designs Or Analyses:**

The overall experimental design seems reasonable. The author compares various baselines across datasets of different scales. The results suggest that for large-scale datasets, the proposed method achieves better performance while maintaining a better convergence speed.

However, in terms of memory efficiency, Table 4 and Table 5 suggest that REST and REST-IS do not show a significant advantage. It would be helpful if the author provided a more in-depth analysis of this aspect, discussing potential trade-offs and explaining why the proposed method does not yield substantial improvements in memory efficiency.

**Methods And Evaluation Criteria:**

N/A

**Other Comments Or Suggestions:**

N/A

**Other Strengths And Weaknesses:**

The paper is well-organized and easy to follow.

The proposed method is simple and can be applied to existing GNNs.

The main concern is that the experimental results do not demonstrate significant improvements in terms of computational complexity. Additionally, the applicability of the proposed method seems restricted to scenarios where the memory table updates at a different frequency than the model.

**Questions For Authors:**

1. In the experiment session, Table 4 and Table 5 suggest that REST and REST-IS do not show a significant advantage. It would be helpful if the author provided a more in-depth analysis of this aspect, discussing potential trade-offs and explaining why the proposed method does not yield substantial improvements in memory efficiency.

2. Do you think the difference in update frequencies is a general issue, or is it specific to historical embeddings? Could your method be extended to other scenarios involving asynchronous updates?

**Relation To Broader Scientific Literature:**

N/A

**Theoretical Claims:**

The given theory looks solid.

---

> ### Author Rebuttal · Authors · 2025-04-01
>
> **A1** Memory Efficiency:
>
> First, we highlight that REST consistently demonstrate significant advantages over GraphSAGE in terms of memory efficiency across all datasets as shown in Table 4 and 5.
>
> When comparing REST with GAS and VR-GCN, REST maintains a similar memory cost. This is because our work aims to introduce a novel, general training framework that can be applied to any historical embedding model to address the serious staleness issue by adjusting the execution frequency of forward propagation, rather than altering the model architecture, thereby naturally preserving the memory efficiency of the chosen baselines. This design choice makes our approach compatible with any historical embedding methods.
>
> We want to emphasis that baseline methods come with more notable drawbacks— deteriorate performance and slow convergence—than memory efficiency. Therefore, our work focuses on resolving these critical issues to benefit all existing methods and provide a promising solution for future research, rather than further reducing memory cost.
>
> **A2** Computational Complexity:
>
> We would like to emphasize that REST still demonstrates significant computational advantages over the baselines. In terms of time complexity, our model consistently achieves much faster convergence and requires considerably less running time compared with SOTA historical embedding baselines such as GAS (Table 4, 12; Figure 5, 6, 10, 11), VR-GCN (Table 5; Figure 12, 13), LMC (Table 7). It maintains a comparable running time to GraphSAGE while achieving superior performance. Regarding memory complexity, please refer to the previous answer. Overall, since related works are more severely impacted by staleness  than by memory cost, our work focuses on addressing critical staleness problems in these methods. REST aims to enhance performance and accelerate convergence while preserving the high scalability that these methods have achieved.
>
> **A3** Applicability of REST:
>
> We would like to clarify that the memory table being updated at a different frequency from the model is a key advantage of our method rather than a limitation. Based on our finding that frequency mismatch is the root cause of staleness, our model is designed to decouple the forward and backward operations, enabling the memory table to be refreshed at a flexible frequency that existing works cannot achieve. In other words, REST actively assigns different update frequencies, thereby alleviating the staleness. If we set the frequency $f$ to 0, it degenerates into the conventional training mode. Thus, our approach is actually an general form of the existing methods rather than an imposed restriction.
>
> In summary, REST is not limited to any specific case; rather, it represents a novel strategy that enables flexible adjustment of the forward and backward operations to address the root cause of staleness. Moreover, it is adaptable to existing training frameworks and broader techniques, making it both highly valuable and widely applicable.
>
> **A4** Broad Impact and asynchronous scenario:
>
> REST addresses the general issue of staleness arising from asynchronous updates—a common challenge in various scenarios, not just limit to the historical embeddings, even though our focus is on large-scale GNN training in this work. We provide several other common scenarios in which REST can be applied:
>
> (1) Distributed Training:
> Asynchronous updates often arise in distributed training due to communication overhead. Nodes may operate on potentially stale global parameters from central servers, leading to parameter staleness. Applying REST in this scenario means performing extra forward passes without gradient calculation asynchronously, which refreshes local embeddings (or activation caches) more frequently. Consequently, when the node eventually computes gradients or synchronizes with the central server, the resulting update is less affected by staleness, improving overall convergence.
>
> (2) Federated Learning:
> Federated learning suffers from parameter staleness due to infrequent client–server communications. REST’s decoupling strategy can be applied by letting the clients (or server) perform additional forward-only steps between global synchronizations. These extra forward passes keep local representations up to date with the client’s current model version, so that when the client finally computes gradients and communicates them back, the cached embeddings are no longer heavily stale. These additional asynchronous forward updates serve as opportunities for refreshing beyond global synchronization. This mitigates the mismatch arising from stale parameters and promotes faster convergence in federated learning.
>
> This broader applicability indicates REST is not merely a method specific to historical embeddings, but a generally beneficial framework for addressing asynchronous update issues widely prevalent in machine learning training. We believe that this idea can offer new insights for various scenarios.

---

### Decision · Program_Chairs · 2025-05-01

**Decision:**

Accept (poster)

**Comment:**

This work aims to reduce feature staleness in GNNs. Staleness occurs when, to scale GNNs, "historical" features are used for some neighboring nodes during the aggregation. After some empirical explorations, the paper proposes different frequencies for forward and backward propagation, and evaluates them empirically.
During the rebuttal and discussion, the relation to prior work was an issue that was discussed. While the relation to GAS was resolved, overall, the reviewers judged the novelty of the work as not too high, which made this a borderline paper. However, what swayed the opinion towards positive was the empirical evaluation with large-scale experiments, which also show the effectiveness of the proposed method in achieving higher accuracy and better convergence.

One main issue that remains is the assumption of the theorems that the gradients are Lipschitz. Why does this hold? Why is the gradient for the true feature and that for the approximation even the same object/function? This must be explained or the theorems toned down as a contribution accordingly, with the assumption discussed.

There are still some modifications that the authors need to make:

(1) The proofs of all theorems must be in the paper to make it self-contained, it is not enough to cite prior work unless that work has proven the exact same result. This also includes specifying and discussing all assumptions (there was no consensus yet on the Lipschitz continuity, and when it may or may not hold; this should be discussed in light of the algorithm).

(2) Some reviewers could not follow all terminology or apparently misunderstood something. The writing should be accessible by the general ICML audience.
I feel Section 2 could also gain in clarity, e.g. the discussion about convergence analysis. It would also be clearer, in addition to the provided plots, to directly check the correlation of memory consistence with accuracy/convergence, instead of going indirectly via batch size.